# UniOD: A Universal Model for Outlier Detection across Diverse Domains

**Dazhi Fu**    **Jicong Fan**[*]
School of Data Science
The Chinese University of Hong Kong, Shenzhen
`dazhifu@link.cuhk.edu.cn, fanjicong@cuhk.edu.cn`

## Abstract

Outlier detection (OD), distinguishing inliers and outliers in completely unlabeled datasets, plays a vital role in science and engineering. Although there have been many insightful OD methods, most of them require troublesome hyperparameter tuning (a challenge in unsupervised learning) and costly model training for every task or dataset. In this work, we propose UniOD, a universal OD framework that leverages labeled datasets to train a single model capable of detecting outliers of datasets with different feature dimensions and heterogeneous feature spaces from diverse domains. Specifically, UniOD extracts uniform and comparable features across different datasets by constructing and factorizing multi-scale point-wise similarity matrices. It then employs graph neural networks to capture comprehensive within-dataset and between-dataset information simultaneously, and formulates outlier detection tasks as node classification tasks. As a result, once the training is complete, UniOD can identify outliers in datasets from diverse domains without any further model/hyperparameter selection and parameter optimization, which greatly improves convenience and accuracy in real applications. More importantly, we provide theoretical guarantees for the effectiveness of UniOD, consistent with our numerical results. We evaluate UniOD on 30 benchmark OD datasets against 17 baselines, demonstrating its effectiveness and superiority. Our code is available at `https://github.com/fudazhiaka/UniOD`.

## 1 Introduction

Outliers are observations that deviate substantially from other normal data in a dataset, indicating they likely arise from a distinct generative process. In data-driven applications, detecting and removing outliers is vital, since their presence can severely degrade the accuracy and robustness of downstream analyses. The problem of identifying such anomalies—commonly termed outlier detection (OD) (Hodge & Austin, 2004; Chandola et al., 2009; Ruff et al., 2021), anomaly detection (Pang et al., 2021), or novelty detection (Pimentel et al., 2014)—has attracted extensive research. OD techniques serve a variety of purposes (Singh & Upadhyaya, 2012; Ahmed et al., 2016; Breier & Branišová, 2017), including preprocessing for supervised learning to eliminate aberrant samples, healthcare diagnostics and monitoring, fraud detection in financial transactions and cybersecurity, and beyond.

In the past few decades, many OD methods have been proposed. Basically, we can divide them into two categories: traditional methods and deep-learning (neural network) based methods. Traditional methods often employ kernel functions (Parzen, 1962; Schölkopf et al., 2001), nearest neighbors (Ramaswamy et al., 2000), and decision trees (Liu et al., 2008), among others, to build their models. For instance, local outlier factor (LOF) (Breunig et al., 2000) compares the local density of an observation to the local densities of its neighbors and identifies observations that have a substantially lower density than their neighbors as outliers. Isolation Forest (Liu et al., 2008) assumes that outliers are more susceptible to isolation than normal observations and uses random decision trees recursively to partition the feature space, where outliers tend to be separated into leaf nodes in far fewer splits. Deep learning based OD methods use neural networks for feature extraction, dimensionality reduction, or other purposes. For instance, DeepSVDD (Ruff et al., 2018b) uses a neural network to project

---

[*]Corresponding author

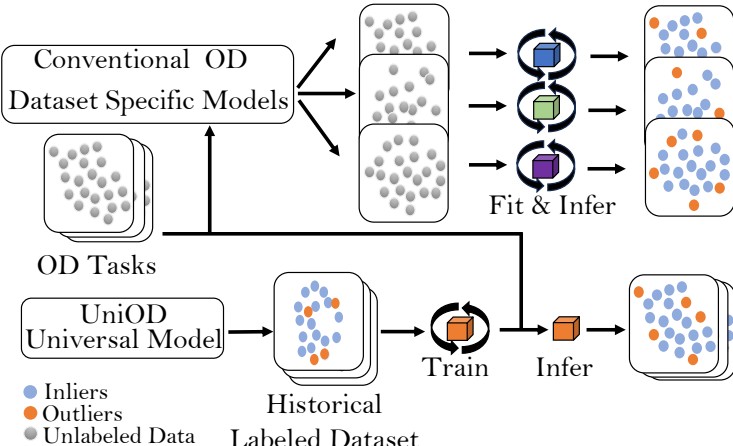

Figure 1: Pipeline comparison between UniOD and conventional OD methods. These approaches train a separate model per dataset, while UniOD leverages a collection of historical labeled datasets to train a single universal model.

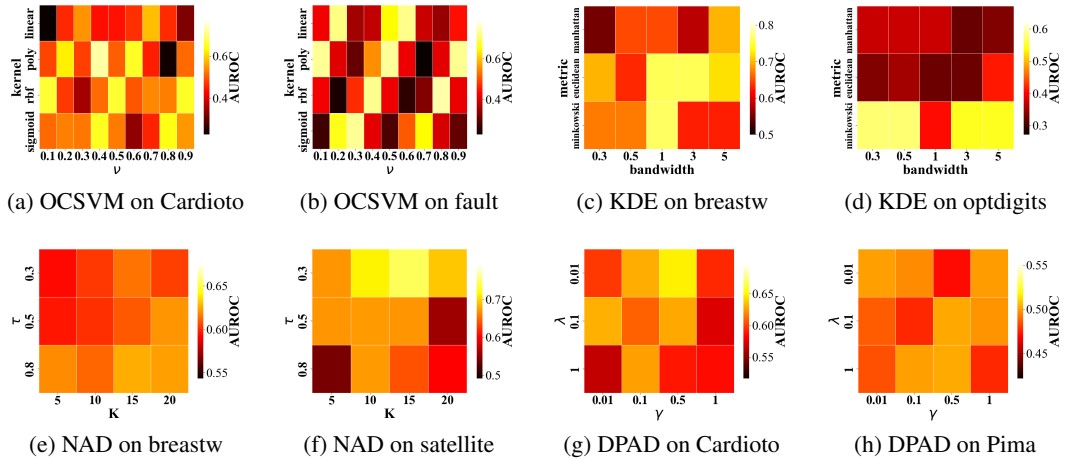

Figure 2: Examples of the performance sensitivity of OD methods to their hyperparameters.

samples into a hypersphere and uses the distance to the center of the hypersphere as an outlier score. NeutralAD (Qiu et al., 2021) uses augmentations on data and maps the augmented data and original data into a space where the embedding of the augmented data remains similar to that of the original data. PLAD (Cai & Fan, 2022) jointly learns a perturbator to perturb each normal data and a classifier to distinguish between the normal data and the perturbed counterpart, where the latter is regarded as a pseudo-abnormal sample. ImAD (Xiao & Fan, 2024) is an end-to-end deep anomaly detection method for data with missing values (Fan, 2025b). Note that there are more deep learning based OD methods (Shenkar & Wolf, 2022; Livernoche et al., 2023a; Xu et al., 2023; Fu et al., 2024; Zhang et al., 2024), and some of them focus on specific data types, such as tabular data (Dai et al., 2025) and graph-structured data (Fan, 2025a; Cai et al., 2025; Wang et al., 2025), or consider special settings such as semi-supervised learning (Ruff et al., 2019; Xiao et al., 2025b) and fairness (Xiao et al., 2025a). Due to space limitations, we will not detail them in this work.

The OD methods mentioned above, especially those based on deep learning, are dataset-specific as shown in Figure 1. That means, for a new dataset, especially when it is from a different domain, we have to train an OD model from scratch, which has the following limitations:

- **High effort on model selection and hyperparameter tuning** Particularly, for deep learning based OD methods, we need to determine the network depth, width, learning rate,

and method-specific hyperparameters. As shown in Figure 2, the optimal hyperparameter combinations vary considerably across different datasets, leading to considerable challenges.

- **High computational cost and waiting time before deployment**   The training or fitting process is often time-consuming, especially when the model and data sizes are large.

- **Waste of knowledge from historical datasets**   Historical datasets often contain useful and transferable knowledge about inlier and outlier patterns, which cannot be effectively used by the conventional OD methods.

In this work, we aim to address the three limitations above by constructing a universal outlier detection model, called UniOD. The main idea is to use labeled historical datasets (widely available) to train a universal model capable of detecting outliers for all other tabular datasets without retraining. Specifically, we extract uniformly dimensioned features for different datasets by building and factorizing multi-scale similarity matrices. We then use graph neural networks to capture comprehensive within-dataset and between-dataset information simultaneously, and formulate outlier detection tasks as node classification tasks, so as to distinguish inliers from outliers. After training, UniOD can be applied to any newly unseen tabular datasets without further hyperparameter tuning and parameter optimization. An overview of UniOD is shown in Figure 3. Our contributions are as follows.

- We propose a novel outlier detection method UniOD, that is able to leverage knowledge from historical datasets and directly classify outliers inside a newly unseen dataset without training.

- UniOD has much lower model complexity compared to other deep learning based methods since outlier detection on all datasets can be done using a single model. Additionally, UniOD is computationally cheaper for outlier detection because it skips retraining.

- We provide theoretical guarantees for the effectiveness of UniOD, which is consistent with the numerical verification.

- We conduct experiments on 30 datasets and compare UniOD against 17 baseline methods, where UniOD outperforms most of them.

## 2   RELATED WORK

**OD Methods without Model Training**   Several nonparametric OD methods, e.g. $k$-nearest neighbor (kNN) (Ramaswamy et al., 2000), avoid explicit training. For instance, kernel density estimation (KDE) (Parzen, 1962) detects low-density data. Local outlier factor (LOF) (Breunig et al., 2000) estimates the local density of each sample and compares local density to that of its neighbors. More recently, ECOD (Li et al., 2022) leverages empirical cumulative distribution functions to capture tail probabilities, highlighting samples that are extreme along one or more feature dimensions.

**Model/Hyperparameter Selection for OD**   OD methods are often sensitive to hyperparameter changes (Goldstein & Uchida, 2016; Ding et al., 2022). To address this, recent approaches leverage prior knowledge from historical datasets for automated hyperparameter or model selection. For example, MetaOD (Zhao et al., 2020) and HPOD (Zhao & Akoglu, 2022) exploit past performance records for prediction, while ROBOD (Ding et al., 2022) ensembles models with different configurations to bypass manual tuning. ELECT (Zhao et al., 2022) incorporates dataset similarity. PyOD2 (Chen et al., 2024) and MetaOOD (Qin et al., 2024) employ large language models to reason about models and datasets. Despite their effectiveness, these methods typically require exhaustive evaluation of hyperparameter combinations on historical datasets, leading to substantial computational cost—especially for deep OD models. Dai & Fan (2025) proposed an inductive anomaly detection approach, which does not apply to the transductive OD setting.

**Transfer Learning for OD**   Transfer learning (Van Haaren et al., 2015; Weiss et al., 2016) has been explored in outlier and anomaly detection to alleviate the scarcity of labeled data by reusing knowledge across related tasks (Andrews et al., 2016; Vercruyssen et al., 2017; Vincent et al., 2020). For instance, LOCIT (Vincent et al., 2020) transfers labeled instances from source to target tasks, and detects anomalies using both unlabeled target and transferred labeled source instances. Although effective in some cases, these methods face two major limitations: (1) they require strong similarity between source and target domains, which is often unmet in practice, especially for heterogeneous

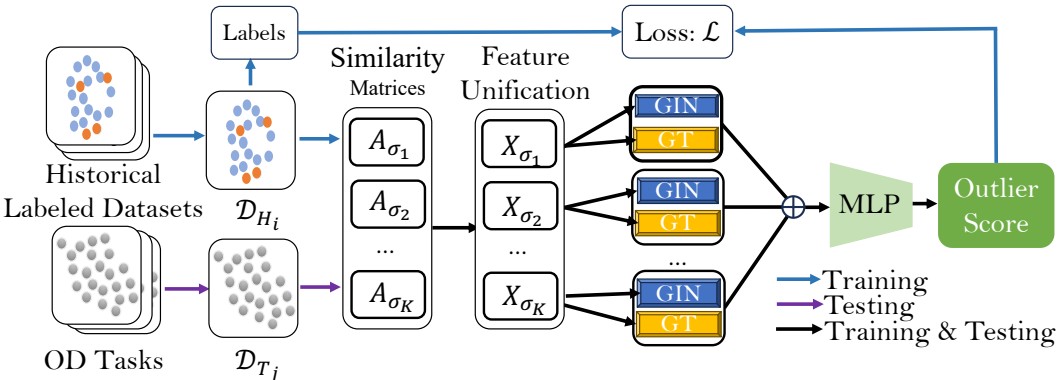

Figure 3: Framework of UniOD. UniOD utilizes multiple labeled datasets to train a universal GNN-based classifier that generalizes across data dimensions and domains for OD.

tabular data; and (2) they require matched feature spaces, excluding the use of source datasets with differing dimensionalities or semantics.

# 3 METHODOLOGY

## 3.1 PROBLEM STATEMENT

Formally, let $T_i$ denote "task $i$", and let $\mathcal{D}_{T_i} = \{\mathbf{x}_{T_i}^{(1)}, \mathbf{x}_{T_i}^{(2)}, \ldots, \mathbf{x}_{T_i}^{(n_{T_i})}\}$ be a set of $d_{T_i}$-dimensional data and assume that it can be partitioned into two subsets, $\mathcal{D}_{T_i}^{\text{inlier}}$ and $\mathcal{D}_{T_i}^{\text{outlier}}$, where $|\mathcal{D}_{T_i}^{\text{inlier}}| \gg |\mathcal{D}_{T_i}^{\text{outlier}}|$ and the data points in $\mathcal{D}_{T_i}^{\text{inlier}}$ are drawn from an unknown distribution with a density function $p_{\text{inlier}}$ such that $p_{\text{inlier}}(\mathbf{x}) > p_{\text{inlier}}(\mathbf{x}')$ holds for any $\mathbf{x} \in \mathcal{D}_{T_i}^{\text{inlier}}$ and $\mathbf{x}' \in \mathcal{D}_{T_i}^{\text{outlier}}$. The primary task of outlier detection (OD) is to identify $\mathcal{D}_{T_i}^{\text{outlier}}$ from $\mathcal{D}_{T_i}$. Notably, suppose there are $B$ datasets, $\mathcal{D}_{T_1}, \mathcal{D}_{T_2}, \ldots, \mathcal{D}_{T_B}$, from $B$ different domains, corresponding to $B$ independent OD tasks, existing OD methods train one model on each dataset independently, which implies $B$ times of hyperparameter tuning and model training or fitting. In addition, the knowledge from datasets across different domains is overlooked. For a new dataset $\mathcal{D}_{T_{B+1}}$, we need to train an OD model from scratch.

## 3.2 PROPOSED METHOD

To tackle the challenges faced by previously proposed OD methods, we propose UniOD, which is able to use labeled historical datasets from different domains, easily available in this big data era, to train a general model that can detect outliers in a dataset from any unseen domain without conducting any retraining. Specifically, let $\mathscr{D}_H = \{\mathcal{D}_{H_1}, \mathcal{D}_{H_2}, \ldots, \mathcal{D}_{H_M}\}$ be a set of $M$ historical labeled datasets, which means:

$$\mathcal{D}_{H_i} = \big\{(\mathbf{x}_{H_i}^{(1)}, \mathbf{y}_{H_i}^{(1)}), (\mathbf{x}_{H_i}^{(2)}, \mathbf{y}_{H_i}^{(2)}), \ldots, (\mathbf{x}_{H_i}^{(n_{H_i})}, \mathbf{y}_{H_i}^{(n_{H_i})})\big\}, \quad i = 1, 2, \ldots, M$$
$$\mathbf{x}_{H_i}^{(j)} \in \mathbb{R}^{d_{H_i}}, \ \mathbf{y}_{H_i}^{(j)} \in \big\{(0,1), (1,0)\big\}, \quad j = 1, 2, \ldots, n_{H_i} \tag{1}$$

where $n_{H_i}$ and $d_{H_i}$ denotes the number and dimension of data in historical dataset $\mathcal{D}_{H_i}$ respectively. $\mathbf{y}_{H_i}^{(j)} = (0, 1)$ indicates that the sample is an outlier. Also, let $\mathscr{D}_T = \{\mathcal{D}_{T_1}, \mathcal{D}_{T_2}, \ldots, \mathcal{D}_{T_B}\}$ be a set of $B$ unlabeled test datasets. The primary goal of UniOD is to train a universal deep-learning based model using these historical datasets $\mathscr{D}_H$, which can directly detect outliers in these test datasets $\mathscr{D}_T$ without any further training or tuning.

### 3.2.1 MULTI-SCALE SIMILARITY-BASED DATA UNIFICATION OF UNIOD

Considering that datasets often differ in dimensionality, feature semantics, and sample size, we first apply preprocessing to harmonize their feature spaces—standardizing both the number of dimensions and the semantic interpretation of each feature. Our key idea is to represent the datasets as point-wise similarity matrix (graph), since a well-designed graph can capture the local and global structures of a dataset, and graph-based methods, such as spectral clustering, provide promising performance

in solving various machine learning problems. Specifically, for each dataset $\mathcal{D}_{H_i}$, we calculate a similarity matrix $A_{H_i,\sigma}$, which induces a weighted graph, using a Gaussian kernel function with hyperparameter $\sigma$, i.e.,

$$\mathbf{A}_{H_i,\sigma}^{(a,b)} = \exp\left(\frac{-\|\mathbf{x}_{H_i}^{(a)} - \mathbf{x}_{H_i}^{(b)}\|^2}{2\sigma^2}\right), \ \forall a,b \in \{1,2,\ldots,n_{H_i}\} \tag{2}$$

Here, we encounter two problems. The first one is how to select an appropriate $\sigma$ for (2). The other is that converting a whole dataset into one single similarity matrix will result in too much loss of information. To solve the two problems, we choose $K$ different $\sigma$, denoted as $\sigma_1, \sigma_2, \ldots, \sigma_K$, and each is determined by $\sigma_k = \beta_k\bar{\sigma}$, where $\beta_k$ takes a value around 1 and $\bar{\sigma}$ is set as the average distance between the data points in $\mathcal{D}_{H_i}$, i.e., $\bar{\sigma} = n_{H_i}^{-2} \sum_{\mathbf{x}\in\mathcal{D}_{H_i}} \sum_{\hat{\mathbf{x}}\in\mathcal{D}_{H_i}} \|\mathbf{x}-\hat{\mathbf{x}}\|$. Consequently, we generate multiple similarity matrices $\mathcal{A}_{H_i} = \{\mathbf{A}_{H_i,\sigma_k}\}_{k=1}^K$ for each dataset. For each similarity matrix $\mathbf{A}_{H_i,\sigma_k}$, we use singular value decomposition (SVD) to generate uniformly dimensioned features across different datasets:

$$\begin{aligned}
\mathbf{A}_{H_i,\sigma_k} &= [\mathbf{u}_1,\mathbf{u}_2,\ldots,\mathbf{u}_{n_{H_i}}]\mathrm{diag}(\lambda_1,\lambda_2,\ldots,\lambda_{n_{H_i}})[\mathbf{v}_1,\mathbf{v}_2,\ldots,\mathbf{v}_{n_{H_i}}]^\top \\
\mathbf{X}_{H_i,\sigma_k} &= [\mathbf{u}_1,\mathbf{u}_2,\ldots,\mathbf{u}_d]\mathrm{diag}(\lambda_1^{1/2},\lambda_2^{1/2},\ldots,\lambda_d^{1/2})
\end{aligned} \tag{3}$$

where $0 < d < n_{H_i}$ is the unified feature dimension. For convenience, we write $\mathbf{X}_{H_i,\sigma_k} = [\tilde{\mathbf{x}}_{H_i,\sigma_k}^{(1)}, \tilde{\mathbf{x}}_{H_i,\sigma_k}^{(2)}, \ldots, \tilde{\mathbf{x}}_{H_i,\sigma_k}^{(n_{H_i})}]^\top \in \mathbb{R}^{n_{H_i}\times d}$, which is the uniformly dimensioned feature matrix of similarity matrices $\mathbf{A}_{H_i,\sigma_k}$, $k = 1,2,\ldots,K$. As a result, for the whole $\mathscr{D}_H = \{\mathcal{D}_{H_1}, \mathcal{D}_{H_2}, \ldots, \mathcal{D}_{H_M}\}$, we obtain

$$\mathscr{X}_H = \{\tilde{\mathbf{X}}_{H_1}, \tilde{\mathbf{X}}_{H_2}, \ldots, \tilde{\mathbf{X}}_{H_M}\} \tag{4}$$

where $\tilde{\mathbf{X}}_{H_i} = (\mathbf{X}_{H_i,\sigma_1}, \ldots, \mathbf{X}_{H_i,\sigma_K}) \in \mathbb{R}^{n_{H_i}\times Kd}$. Similarly, for the whole $\mathscr{D}_T = \{\mathcal{D}_{T_1}, \mathcal{D}_{T_2}, \ldots, \mathcal{D}_{T_B}\}$, we have

$$\mathscr{X}_T = \{\tilde{\mathbf{X}}_{T_1}, \tilde{\mathbf{X}}_{T_2}, \ldots, \tilde{\mathbf{X}}_{T_B}\} \tag{5}$$

The role of converting each dataset into multiple similarity matrices and extracting SVD-based features is twofold. First, constructing similarity matrices and applying SVD embeddings removes dependence on the original feature dimensionality and semantics, thereby enabling a unified representation across heterogeneous datasets. Second, by applying a simple subsampling technique (introduced later) to each historical dataset, we can generate a diverse collection of training data for UniOD, which substantially enhances its generalization capability.

### 3.2.2 MODEL DESIGN OF UNIOD

After generating uniformly dimensioned features $\mathscr{X}_H$, one straightforward option is to train a classifier with an MLP to detect outliers in $\mathscr{D}_H$. However, this approach discards valuable information contained in the similarity matrices $\mathbf{A}_{H_i,\sigma_k}$. To fully exploit such point-wise similarity information, we treat each dataset as a collection of graph-structured data, where the adjacency matrices $\{\mathbf{A}_{H_i,\sigma_k}\}_{k=1}^K$ define the graph structure and $\{\mathbf{X}_{H_i,\sigma_k}\}_{k=1}^K$ define the corresponding node features. In this way, the outlier detection tasks for the historical datasets $\mathscr{D}_H$ are reformulated as binary node classifications over graph-structured data. To deal with graph-structured data, we use $K$ graph isomorphism networks (GINs) (Xu et al., 2018) and $K$ graph transformers (GTs) to build our model, where each GIN is composed of $L_1$ layers and each GT is composed of $L_2$ layers. Specifically, we let

$$\mathbf{Z}_{H_i}^{\mathrm{GIN}} = \mathrm{KGIN}_{\theta_1}(\tilde{\mathbf{X}}_{H_i}, \mathcal{A}_{H_i}), \quad i = 1,2\ldots,M \tag{6}$$

$$\mathbf{Z}_{H_i}^{\mathrm{GT}} = \mathrm{KGT}_{\theta_2}(\tilde{\mathbf{X}}_{H_i}), \quad i = 1,2,\ldots,M \tag{7}$$

where $\theta_1$ and $\theta_2$ denote the parameters of the GINs and GTs, respectively. The details are provided in Appendix C. Then the final embedding is the following concatenation

$$\mathbf{Z}_{H_i} = \left[\mathbf{Z}_{H_i}^{\mathrm{GIN}}, \mathbf{Z}_{H_i}^{\mathrm{GT}}\right] \in \mathbb{R}^{n_{H_i}\times\hat{d}} \tag{8}$$

where $\hat{d}$ is the dimension of final embedding. Then we apply an $L_3$-layer MLP with parameters $\theta_3$ followed by a softmax function to predict the labels:

$$\hat{\mathbf{Y}}_{H_i} = \mathrm{softmax}\left(\mathrm{MLP}_{\theta_3}(\mathbf{Z}_{H_i})\right), \quad i = 1,2,\ldots,M \tag{9}$$

where $\mathrm{MLP}_{\theta_3} : \mathbb{R}^{\hat{d}} \to \mathbb{R}^2$. Then, we minimize cross entropy loss $\mathcal{L}(\theta)$ to train our proposed UniOD:

$$\mathcal{L}(\theta) = -\frac{1}{M} \sum_{i=1}^{M} \sum_{j=1}^{n_{H_i}} \left\langle \mathbf{y}_{H_i}^{(j)}, \log \hat{\mathbf{y}}_{H_i}^{(j)} \right\rangle \tag{10}$$

where $\theta = \{\theta_1, \theta_2, \theta_3\}$ denote all parameters of our model.

After training, for each specific testing dataset $\mathcal{D}_{T_i} \in \mathscr{D}_T$, we can use (2) and (3) to construct multiple graph-structured data and obtain $\tilde{\mathbf{X}}_{T_i}, \mathcal{A}_{T_i}$, and then we can input these graphs into our trained GINs and GTs to obtain $\mathbf{Z}_{T_i}$, and finally use the trained classifier $\mathrm{MLP}_{\theta_3^*}$ to obtain the outlier score for each data point:

$$Outlier\ Score\big(\mathbf{x}_{T_i}^{(j)}\big) = \big[\hat{\mathbf{y}}_{T_i}^{(j)}\big]_2 \tag{11}$$

where $[\hat{\mathbf{y}}_{T_i}^{(j)}]_2$ denotes the second element of a vector $\hat{\mathbf{y}}_{T_i}^{(j)}$. In UniOD, a larger outlier score indicates a higher probability for a data point to be an outlier.

In summary, for each historical dataset (1), we construct multiple graph-structured representations via (2) and (3), and reformulate the outlier-detection problem as a node-classification task. These graphs are then used to train our GNN-based classifier (6,7 and 9) using (10). After training, any new dataset is processed through the same graph-construction pipeline and fed into the pretrained model to assign outlier scores according to (11), thereby identifying outliers in the newly unseen datasets.

### 3.2.3 Algorithm and Implementation of UniOD

We provide a detailed algorithm including both the training and testing stages of UniOD in Algorithm 1, which is in Appendix C due to space limitations. Note that to enhance the generalization capability of UniOD, we create 5 synthetic datasets by randomly subsampling $60\%$ samples from each $\mathcal{D}_{H_i}$, where the outlier (anomaly) ratio remains unchanged from the original dataset. This operation is denoted as $Subsampling(\mathcal{D}_{H_i})$ in Algorithm 1.

## 4 Theoretical Analysis

### 4.1 Time Complexity Analysis

We compare the time complexity of detecting outliers in a new dataset $\mathcal{D}_T$ (with $n$ samples) between the proposed UniOD and other deep-learning methods. To simplify the comparison, all neural networks are assumed to share the same maximum hidden dimension $\bar{d}$. For the deep-learning baselines (excluding UniOD), we fix the number of training epochs to $Q$ and the MLP depth to $L$. For UniOD, we denote its total MLP layers by $\bar{L}$ and its attention layers by $\bar{L}'$. Table 1 shows the comparison among UniOD, ICL, and DPAD, while the complete comparison for all deep learning methods is in Appendix D.

Table 1: Time complexity comparison (partial)

|       | Time Complexity |
|-------|-----------------|
| DPAD  | $\mathcal{O}\big(Qn\bar{d}(L\bar{d}+n)\big)$ |
| ICL   | $\mathcal{O}\big(QndL\bar{d}^2\big)$ |
| UniOD | $\mathcal{O}\big(n^2(d+K\bar{d}) + n\bar{d}^2\bar{L} + n^2\bar{d}\bar{L}'\big)$ |

Previously proposed deep learning OD methods require training for each specific dataset, resulting in an increasing training time as the number of datasets grows. In contrast, UniOD uses only one model for all datasets, and its training phase is entirely decoupled from $\mathcal{D}_T$, eliminating any need for per-dataset retraining. This decoupling enables UniOD to perform online outlier detection, yielding greater efficiency compared to methods that require both training and testing stages for each dataset.

Also, we compare the number of hyperparameters of different deep learning methods in Appendix E.

### 4.2 Generalization Ability Analysis

In this section, we provide a theoretical guarantee for the effectiveness of the proposed model. Particularly, we analyze that once the model is well-trained, it is able to provide high detection accuracy on unseen datasets. This is nontrivial due to the following challenges:

- Given that we are training a universal model for outlier detection across diverse domains, the training data is not a single dataset. Instead, it is composed of multiple different datasets, which makes the analysis more complex than the traditional analysis on a single dataset.

- Because of the graph construction and SVD embedding (see (2) and (3)), in each dataset, the data points are no longer mutually independent, which makes some classical tools in learning theory inapplicable.

- The model consists of $K$ GINs, $K$ GTs, and one MLP, making it a very complex structure, which complicates the theoretical analysis.

For convenience, we use $\mathcal{W}$ to denote the set of all weight matrices of our deep learning model $f$, and use $\|\cdot\|_F$, $\|\cdot\|_2$, and $\|\cdot\|_{2,1}$ to denote the Frobenius norm, spectral norm (largest signular value), and $\ell_{2,1}$ norm (sum of $\ell_2$ norms of columns) of matrix, respectively. To simplify the notation and analysis, without loss of generality, we let: 1) $n_{T_1} = n_{T_2} = \cdots n_{T_M} = n$; 2) All GINs and all GTs have a common layer number $L$, all MLPs (including those in GINs and GTs and the one following their concatenation) have a common depth $L'$, and every multi-heads attentions in the GTs have $H$ heads; 3) In GTs, the layer normalization and residual connection are omitted since they have tiny impact on the derivation and result; 4) $\bar{d}$ is the maximum of the widths of all layers in $f$. Although in (10) we used the cross-entropy loss, other loss functions, such as mean square error, mean absolute error, and hinge loss, are also applicable. Therefore, for theoretical analysis, we consider a general loss function denoted as $\ell(\mathbf{y}, \hat{\mathbf{y}})$. Then the average training error (empirical risk) on the $M$ historical datasets is denoted as $\hat{\mathcal{L}}(f) := \frac{1}{M} \sum_{i=1}^M \frac{1}{n} \sum_{j=1}^n \ell(\mathbf{y}_{H_i}^{(j)}, \hat{\mathbf{y}}_{H_i}^{(j)})$. The expected test error (true risk) is denoted as $\mathcal{L}(f) := \mathbb{E}[\frac{1}{|\mathbf{y}|} \sum_{j=1}^{|\mathbf{y}|} \ell(\mathbf{y}^{(j)}, \hat{\mathbf{y}}^{(j)})]$, where $|\mathbf{y}|$ denotes the number of samples in $\mathbf{y}$. We are going to analyze the gap between $\mathcal{L}(f)$ and $\hat{\mathcal{L}}(f)$. Note that if we let $\ell$ be $\mathbb{1}(\mathbf{y} \neq \hat{\mathbf{y}})$, $1 - \mathcal{L}(f)$ will be the expected detection accuracy on test datasets.

**Theorem 4.1.** *Let* $b_A = \max_{i,k} \|\mathbf{A}_{H_i, \sigma_k}\|_2$, $c_X = \max_k \sqrt{\sum_i \|\mathbf{X}_{H_i, \sigma_k}\|_F^2}$, $b_W = \max_{\mathbf{W} \in \mathcal{W}} \|\mathbf{W}\|_2$, $b'_W = \max_{\mathbf{W} \in \mathcal{W}} \|\mathbf{W}\|_{2,1}$. *Suppose all activation functions are* 1-*Lipschitz, and* $\ell$ *is* $\mu$-*Lipschitz and bounded by* $\beta$. *Then, with probability at least* $1 - \delta$ *over the randomness of* $\mathscr{D}_H$, *the following inequality holds*

$$\mathcal{L}(f) \leq \hat{\mathcal{L}}(f) + \frac{8\beta\sqrt{n} + 24\sqrt{K}\mu b_W^{L'}(C_{GIN} + C_{GT})\ln(M)}{M\sqrt{n}} + 3\beta\sqrt{\frac{\ln(2/\delta)}{2M}} \tag{12}$$

*where* $C_{GIN} = b_A^L c_X b_W^{LL'} L^{3/2} L'^{3/2} (b'_W/b_W) \sqrt{\ln(2\bar{d}^2)}$, $C_{GT} = dc_X H^{L/2} b_W^{L(1+L')} \prod_{i=1}^L s^{(i-1)} \sqrt{\ln(2d^2)}$, *and* $s^{(i-1)} = b_W^2 b_Z^{(i-1)} \bar{d}^{-1/2} + \sqrt{n}$. *Particularly, assuming* $\eta$-*Lipschitz for self-attentions yields* $C_{GT} = dc_X b_W^{LL'} H^{L/2} \eta^L \sqrt{\ln(2d^2)}$.

The proof of the theorem is in Appendix A. Note that $b_Z^{(i-1)}$ is too complex and is detailed in the proof. The assumptions made in the theorem are mild. For example, regarding the loss functions, ReLU, Sigmoid, and Tanh are all 1-Lipschitz continuous. The theorem has the following important implications:

- When there are more training datasets, namely, $M$ is larger, the bound is tighter. This is consistent with our numerical result shown by Figure 4a.

- Due to the $\sqrt{K}$ in the bound, increasing the number $K$ of parallelized GINs and GTs has a tiny impact on the gap between $\mathcal{L}(f)$ and $\hat{\mathcal{L}}(f)$ but it can reduce the training error, thereby increasing the testing accuracy. This is consistent with the results shown by Figure 4b.

- When the layer numbers $L$ and $L'$ are too large, the generalization ability of UniOD is weak, especially when the spectral norms of the weight matrices are larger than 1. However, we can use spectral normalization to ensure the small $b_W$ in real applications.

## 5 NUMERICAL RESULTS

### 5.1 EXPERIMENTAL SETTINGS

**Datasets** Our experiments are conducted on ADBench (Han et al., 2022), which is a popular benchmark in outlier (anomaly) detection, containing widely used real-world datasets in multiple

domains, including healthcare, audio, language processing, and finance. Detailed descriptions and statistical information about these datasets are provided in the Appendix F.

**Baselines and Hyperparameter Settings** UniOD is compared with 17 widely-used baseline methods, including traditional methods: KDE (Parzen, 1962), $k$NN (Ramaswamy et al., 2000), LOF (Breunig et al., 2000), OC-SVM (Schölkopf et al., 2001), IF (Liu et al., 2008), LODA (Pevnỳ, 2016), ECOD (Li et al., 2022), deep-learning based methods: AE (Hinton & Salakhutdinov, 2006), DSVDD (Ruff et al., 2018a), NeutralAD (Qiu et al., 2021), ICL (Shenkar & Wolf, 2022), SLAD (Xu et al., 2023), DTE-NP (Livernoche et al., 2023b), DPAD (Fu et al., 2024), KPCA+MLP (Schölkopf et al., 1998), MLP+TF, and one model selection method: MetaOD (Zhao et al., 2020). It is noteworthy that methods such as ELECT (Zhao et al., 2022) and HPOD (Zhao & Akoglu, 2022) are not included, since their meta-feature construction processes are not publicly available. MetaOOD (Qin et al., 2024) is designed for out-of-distribution detection, and AutoUAD (Dai & Fan, 2025) is limited to inductive anomaly detection. For DTE-NP, DPAD, SLAD, ICL, and NeutralAD, we use the code provided by the authors. As for other methods, we use the code from the Python library PyOD (Zhao et al., 2019). For both traditional and deep methods, we use grid search to obtain the best-performing set of hyperparameters in the historical datasets, which is then used in our experiments. The detailed hyperparameter configuration for grid search and MetaOD is provided in Appendix G. KPCA+MLP and MLP+TF are two baseline methods that uses historical labeled datasets for training. More details are provided in Appendix H.1.

**Implementation** In our experiments, we consider 30 datasets and partition them into two equal groups (Group I and Group II). We use Group II as the historical datasets to train UniOD and use Group I as the testing datasets. We also exchange the roles of the two sets. More implementation details are provided in Appendix H.2.

**Performance Metrics** We use two metrics to evaluate the performance of UniOD and other baseline methods: Area Under the Receiver Operating Characteristic Curve (AUROC) and the Area Under the Precision-Recall Curve (AUPRC), following (Xu et al., 2023; Livernoche et al., 2023b; Kim et al., 2024). The two metrics are threshold-free and hence can avoid the uncertainty and unfairness in determining the thresholds for the OD methods.

Table 2: Average AUROC (%) and AUPRC (%) of each method over 5 runs, including both training and testing, on Group I datasets. The best and second-best results are highlighted in red and orange, respectively.

| AUROC | KDE (1962) | kNN (2000) | AE (2006) | DSVDD (2018) | NeutralAD (2021) | ECOD (2022) | ICL (2022) | SLAD (2023) | DPAD (2024) | DTE-NP (2024) | KPCA+MLP | MLP+TF | MetaOD (2021) | UniOD Ours |
|---|---|---|---|---|---|---|---|---|---|---|---|---|---|---|
| breastw | 98.43 | 98.47 | 95.12 | 82.88 | 84.92 | 99.14 | 82.57 | 81.35 | 86.29 | 97.89 | 88.16 | 97.21 | 97.71 | 99.10 |
| Cardio. | 50.27 | 51.91 | 55.55 | 71.36 | 38.63 | 78.53 | 40.82 | 32.60 | 30.62 | 49.23 | 97.21 | 74.41 | 60.51 | 51.20 |
| fault | 73.05 | 71.46 | 66.25 | 48.94 | 67.35 | 46.87 | 55.29 | 71.83 | 66.77 | 73.44 | 90.55 | 53.00 | 57.21 | 69.60 |
| InternetAds | 61.79 | 62.36 | 53.71 | 62.18 | 66.42 | 67.70 | 42.73 | 64.66 | 65.11 | 58.18 | 54.47 | 69.62 |  | 63.50 |
| landsat | 62.46 | 61.58 | 49.87 | 42.59 | 61.92 | 36.78 | 69.91 | 67.24 | 52.72 | 59.20 | 39.04 | 40.08 | 56.80 | 69.10 |
| optdigits | 32.32 | 39.91 | 45.11 | 40.74 | 69.38 | 60.45 | 54.18 | 55.29 | 50.63 | 36.38 | 56.63 | 44.04 | 87.22 | 73.80 |
| PageBlocks | 90.66 | 92.08 | 91.63 | 89.72 | 84.17 | 91.39 | 62.34 | 74.08 | 52.70 | 89.54 | 76.26 | 71.41 | 75.99 | 88.20 |
| pendigits | 89.05 | 90.23 | 79.46 | 87.87 | 77.51 | 92.74 | 36.64 | 60.11 | 63.73 | 77.08 | 64.61 | 85.45 | 72.11 | 77.50 |
| Pima | 72.28 | 72.53 | 63.86 | 63.21 | 61.19 | 59.44 | 51.15 | 47.75 | 61.46 | 71.54 | 63.17 | 68.76 | 70.89 | 72.30 |
| satellite | 76.03 | 73.13 | 68.52 | 57.25 | 62.32 | 58.30 | 59.07 | 68.21 | 64.76 | 68.91 | 55.30 | 61.50 | 65.20 | 86.90 |
| satimage-2 | 96.44 | 99.88 | 95.71 | 96.68 | 79.71 | 96.49 | 64.50 | 95.97 | 81.22 | 96.59 | 59.62 | 96.74 | 91.51 | 99.70 |
| SpamBase | 49.52 | 56.14 | 55.99 | 56.82 | 51.09 | 65.56 | 21.68 | 52.55 | 47.14 | 53.14 | 53.39 | 51.36 | 66.20 | 56.30 |
| thyroid | 95.83 | 96.49 | 96.27 | 91.35 | 60.25 | 97.71 | 42.61 | 85.12 | 73.53 | 96.37 | 82.68 | 90.96 | 95.01 | 94.10 |
| Waveform | 75.12 | 75.35 | 60.25 | 64.70 | 69.31 | 60.35 | 62.36 | 44.54 | 55.22 | 73.89 | 62.04 | 54.44 | 69.40 | 84.30 |
| WDBC | 95.01 | 98.43 | 84.37 | 97.00 | 30.17 | 97.06 | 86.33 | 70.20 | 93.63 | 97.97 | 89.10 | 94.86 | 96.30 | 98.40 |
| Average | 74.55 | 76.00 | 70.78 | 70.22 | 64.29 | 73.90 | 55.48 | 64.77 | 62.29 | 73.75 | 66.13 | 69.24 | 75.45 | 78.93 |

| AUPRC | KDE (1962) | kNN (2000) | AE (2006) | DSVDD (2018) | NeutralAD (2021) | ECOD (2022) | ICL (2022) | SLAD (2023) | DPAD (2024) | DTE-NP (2024) | KPCA+MLP | MLP+TF | MetaOD (2021) | UniOD Ours |
|---|---|---|---|---|---|---|---|---|---|---|---|---|---|---|
| breastw | 95.58 | 95.64 | 87.90 | 83.10 | 61.29 | 98.39 | 71.74 | 69.59 | 79.45 | 93.03 | 47.85 | 93.53 | 93.54 | 96.90 |
| Cardio. | 27.54 | 33.64 | 30.86 | 43.89 | 20.17 | 50.54 | 18.55 | 21.78 | 16.69 | 31.00 | 29.47 | 52.40 | 36.96 | 35.30 |
| fault | 54.57 | 52.08 | 48.16 | 33.39 | 47.89 | 32.57 | 39.29 | 53.04 | 37.77 | 39.39 | 84.08 | 36.42 | 42.77 | 49.10 |
| InternetAds | 23.80 | 29.76 | 19.33 | 29.83 | 34.68 | 50.89 | 16.07 | 29.35 | 21.11 | 29.39 | 28.39 | 26.59 | 52.50 | 32.80 |
| landsat | 26.03 | 25.72 | 20.23 | 19.02 | 28.33 | 16.35 | 42.64 | 29.21 | 22.14 | 25.13 | 15.95 | 16.84 | 24.29 | 28.10 |
| optdigits | 1.97 | 2.19 | 2.43 | 2.35 | 5.13 | 3.37 | 2.91 | 2.95 | 2.09 | 3.09 | 2.36 |  | 19.74 | 5.00 |
| PageBlocks | 53.98 | 56.83 | 52.12 | 51.98 | 32.69 | 51.99 | 31.58 | 36.85 | 11.82 | 51.11 | 30.61 | 34.99 | 29.08 | 46.20 |
| pendigits | 12.11 | 13.66 | 6.70 | 14.47 | 5.14 | 26.56 | 1.63 | 2.75 | 5.68 | 8.05 | 2.89 | 15.39 | 7.02 | 6.00 |
| Pima | 53.49 | 52.64 | 45.43 | 46.13 | 41.15 | 46.42 | 35.63 | 34.64 | 45.46 | 52.07 | 46.80 | 50.01 | 57.18 | 52.90 |
| satellite | 60.35 | 59.28 | 54.40 | 51.63 | 40.74 | 52.61 | 43.85 | 48.81 | 47.16 | 54.72 | 41.56 | 55.99 | 50.22 | 79.20 |
| satimage-2 | 31.80 | 94.52 | 28.47 | 78.87 | 2.85 | 65.97 | 1.59 | 37.66 | 8.42 | 40.75 | 43.99 | 86.93 | 29.83 | 95.70 |
| SpamBase | 38.29 | 41.36 | 41.91 | 43.23 | 38.63 | 51.83 | 26.72 | 40.66 | 38.82 | 40.38 | 40.41 | 39.71 | 51.20 | 42.50 |
| thyroid | 28.60 | 40.22 | 35.72 | 26.02 | 4.22 | 46.78 | 2.16 | 27.23 | 14.58 | 32.36 | 50.68 | 40.81 | 49.56 | 44.30 |
| Waveform | 11.44 | 13.26 | 4.25 | 4.61 | 33.56 | 4.05 | 6.19 | 2.41 | 4.22 | 11.35 | 5.78 | 3.79 | 4.84 | 9.60 |
| WDBC | 25.49 | 53.81 | 16.87 | 53.35 | 2.02 | 50.53 | 9.73 | 8.61 | 54.07 | 47.17 | 48.51 | 22.88 | 31.99 | 57.80 |
| Average | 36.34 | 44.31 | 32.99 | 38.79 | 26.55 | 43.26 | 23.35 | 29.70 | 28.03 | 38.16 | 34.67 | 38.57 | 38.71 | 45.43 |

Table 3: Average AUROC (%) and AUPRC (%) of each method over 5 runs, including both training and testing, on Group II datasets. The best and second-best results are highlighted in red and orange, respectively.

| AUROC | KDE (1962) | kNN (2000) | AE (2006) | DSVDD (2018) | NeutralAD (2021) | ECOD (2022) | ICL (2022) | SLAD (2023) | DPAD (2024) | DTE-NP (2024) | KPCA +MLP | MLP +TF | MetaOD (2021) | UniOD Ours |
|---|---|---|---|---|---|---|---|---|---|---|---|---|---|---|
| ALOI | 53.35 | 53.57 | 55.00 | 54.85 | 56.16 | 51.60 | 52.11 | 52.73 | 49.36 | 56.72 | 52.86 | 50.82 | 48.38 | 54.39 |
| campaign | 74.19 | 74.21 | 71.54 | 67.28 | 69.06 | 76.24 | 71.41 | 70.22 | 49.79 | 74.28 | 46.46 | 36.08 | 76.21 | 73.23 |
| cardio | 83.90 | 90.99 | 89.81 | 88.94 | 47.68 | 93.50 | 26.99 | 47.82 | 73.23 | 74.34 | 66.37 | 79.59 | 56.44 | 93.77 |
| celeba | 74.70 | 79.71 | 73.67 | 67.28 | 54.49 | 75.25 | 64.90 | 66.13 | 51.73 | 74.60 | 58.34 | 39.47 | 69.73 | 81.21 |
| cover | 87.00 | 91.40 | 89.32 | 89.45 | 73.72 | 89.64 | 54.04 | 66.15 | 69.82 | 91.63 | 62.69 | 77.59 | 88.82 | 93.52 |
| wilt | 34.02 | 44.40 | 44.98 | 31.93 | 52.72 | 39.40 | 57.57 | 59.83 | 51.08 | 58.00 | 57.32 | 32.03 | 51.72 | 52.61 |
| http | 99.57 | 99.57 | 99.87 | 99.70 | 91.71 | 98.27 | 35.05 | 99.95 | 51.61 | 12.54 | 99.90 | 65.70 | 99.95 | 100.00 |
| magic_g. | 70.00 | 76.39 | 72.80 | 67.59 | 65.63 | 64.78 | 65.82 | 63.91 | 55.39 | 80.60 | 73.48 | 63.95 | 70.99 | 70.10 |
| mammog. | 87.17 | 84.92 | 76.52 | 86.95 | 62.21 | 89.68 | 56.47 | 59.06 | 61.99 | 84.60 | 79.12 | 85.23 | 84.77 | 87.25 |
| shuttle | 99.37 | 92.06 | 99.05 | 99.26 | 73.58 | 99.40 | 52.21 | 94.27 | 56.66 | 79.58 | 80.61 | 95.85 | 49.17 | 99.51 |
| skin | 51.25 | 76.04 | 49.55 | 59.51 | 78.65 | 48.86 | 33.57 | 83.12 | 64.64 | 71.43 | 57.39 | 62.75 | 65.19 | 76.12 |
| speech | 52.02 | 47.83 | 47.49 | 45.70 | 53.61 | 46.97 | 43.22 | 52.41 | 50.98 | 49.99 | 48.30 | 45.96 | 54.88 | 46.97 |
| smtp | 100.00 | 100.00 | 99.98 | 100.00 | 99.69 | 100.00 | 99.94 | 100.00 | 74.25 | 100.00 | 99.99 | 78.85 | 97.81 | 100.00 |
| letter | 87.60 | 74.44 | 63.10 | 57.46 | 88.26 | 57.23 | 73.05 | 88.40 | 58.20 | 88.16 | 55.51 | 54.35 | 90.09 | 64.05 |
| vowels | 82.91 | 91.23 | 76.25 | 41.54 | 97.64 | 59.29 | 82.17 | 93.65 | 72.66 | 97.37 | 70.29 | 61.56 | 94.99 | 85.01 |
| Average | 75.80 | 78.45 | 73.93 | 70.50 | 70.99 | 72.67 | 57.90 | 73.18 | 59.43 | 72.92 | 67.24 | 61.99 | 73.28 | 78.52 |
| **AUPRC** | | | | | | | | | | | | | | |
| ALOI | 4.26 | 3.97 | 4.26 | 3.96 | 4.38 | 3.30 | 3.69 | 3.79 | 3.03 | 4.76 | 3.41 | 3.09 | 2.79 | 3.62 |
| campaign | 28.03 | 28.49 | 25.03 | 24.55 | 20.93 | 34.27 | 22.88 | 23.95 | 11.35 | 28.82 | 10.89 | 12.64 | 36.70 | 28.47 |
| cardio | 36.47 | 51.18 | 43.13 | 46.35 | 11.93 | 56.74 | 6.14 | 19.69 | 30.85 | 35.49 | 23.86 | 38.04 | 18.95 | 57.63 |
| celeba | 5.86 | 9.27 | 5.59 | 6.24 | 2.31 | 10.76 | 4.99 | 5.29 | 2.37 | 6.34 | 3.63 | 4.21 | 5.37 | 11.71 |
| cover | 5.17 | 7.77 | 11.16 | 15.60 | 3.85 | 11.97 | 1.04 | 1.46 | 3.07 | 9.59 | 3.94 | 1.94 | 9.64 | 5.89 |
| wilt | 3.67 | 4.34 | 4.42 | 3.61 | 13.01 | 4.17 | 6.05 | 6.29 | 5.93 | 5.73 | 6.01 | 3.58 | 4.98 | 3.67 |
| http | 44.32 | 44.32 | 70.08 | 59.04 | 3.88 | 15.95 | 0.61 | 86.79 | 1.80 | 1.13 | 79.31 | 1.02 | 96.56 | 100.00 |
| magic_g. | 63.52 | 69.83 | 64.36 | 56.21 | 50.11 | 54.50 | 55.26 | 52.45 | 40.51 | 73.46 | 64.26 | 52.87 | 62.54 | 62.48 |
| mammog. | 20.98 | 15.98 | 12.14 | 21.19 | 3.05 | 41.16 | 3.95 | 4.10 | 6.07 | 16.69 | 26.42 | 10.41 | 21.87 | 19.50 |
| shuttle | 91.86 | 34.51 | 79.74 | 91.32 | 14.77 | 91.10 | 8.29 | 39.86 | 12.99 | 19.76 | 16.82 | 54.14 | 8.88 | 96.18 |
| skin | 19.08 | 31.60 | 19.47 | 24.43 | 33.92 | 18.24 | 14.92 | 46.87 | 24.57 | 28.48 | 23.99 | 35.42 | 24.53 | 31.65 |
| speech | 1.77 | 1.88 | 1.84 | 1.99 | 2.28 | 1.96 | 1.39 | 2.48 | 2.08 | 2.01 | 1.47 | 1.96 | 3.87 | 1.84 |
| smtp | 100.00 | 100.00 | 58.33 | 100.00 | 27.63 | 100.00 | 61.11 | 100.00 | 14.09 | 100.00 | 83.33 | 0.22 | 1.12 | 100.00 |
| letter | 35.42 | 15.30 | 11.30 | 9.44 | 42.13 | 7.67 | 17.96 | 36.87 | 8.99 | 30.41 | 6.64 | 9.32 | 52.40 | 10.54 |
| vowels | 23.17 | 30.29 | 21.48 | 4.02 | 56.95 | 8.14 | 27.68 | 33.96 | 18.11 | 56.06 | 5.81 | 6.44 | 35.61 | 17.23 |
| Average | 32.24 | 29.92 | 28.82 | 31.20 | 19.41 | 30.66 | 15.73 | 30.92 | 12.39 | 27.92 | 23.99 | 15.69 | 25.72 | 36.69 |

## 5.2 RESULTS OF OD

The average AUROC and AUPRC performance on 15 datasets from Group I is shown in Table 2. In AUROC and AUPRC, UniOD achieves the best average performance, which is 3% higher than the second-best method-kNN. Compared with KPCA+MLP and MLP+TF, which also use historical datasets, UniOD significantly outperforms them.

Another interesting phenomenon is that simple traditional methods, such as kNN and KDE, outperform most deep learning methods in various datasets. One reason is that in tabular data with low-dimensional features, even a simple Euclidean distance reflects semantic differences between different samples. However, as the dimension of datasets increases, deep-learning based methods will be more powerful since methods like kNN and KDE provide implicit prediction results for high-dimensional datasets as discussed in (Jiang, 2017; Gu et al., 2019).

Table 4: Time costs (seconds) comparison of different deep methods on 15 datasets.

| AE (2006) | DSVDD (2018) | NeutralAD (2021) | ICL (2022) | SLAD (2023) | DPAD (2024) | **UniOD** Ours |
|---|---|---|---|---|---|---|
| 384 | 511 | 664 | 1391 | 485 | 788 | 240 |

To mitigate the influence of specific historical datasets, we conduct cross-validation experiments. In this setting, the 15 datasets in Group I are treated as historical datasets, and evaluation is performed on datasets from Group II. For datasets containing more than 6,000 samples, we subsample them to 6,000 while preserving the original anomaly ratio, due to computational resource constraints. As reported in Table 3, the superior performance of UniOD demonstrates its robustness and effectiveness, indicating that its performance does not depend on specific historical datasets. Meanwhile, we provide TSNE visualization results of the learned representations $\mathbf{Z}_{T_i}$ on several datasets in Figure 5 and observe that most outliers tend to gather into a small, dense cluster while a smaller portion of outliers appear as isolated nodes.

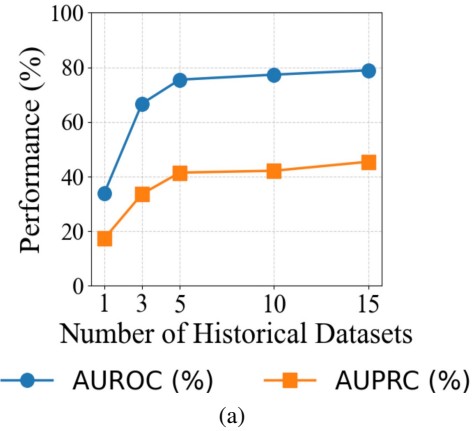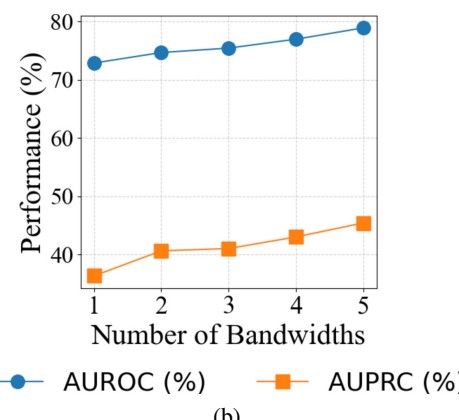

Figure 4: Ablation study: (a) Average performance using different numbers of historical datasets; (b) Average performance using different number of bandwidths for similarity matrices.

Also, we compare the time costs of different deep learning OD methods in detecting the outliers on the 15 datasets from Group I, the results are shown in Table 4. Note that the reported results exclude hyperparameter tuning time. UniOD achieves a lower time cost than those dataset-specific deep OD methods.

## 5.3 ABLATION STUDIES

In this subsection, we conduct experiments to investigate how each component of our proposed UniOD affects its outlier detection performance. We first evaluate the performance of UniOD with $1, 3, 5, 10, 15$ training historical datasets (with sub-sampling augmentation) shown in Figure 4a. It is obvious that as the number of historical datasets expands, its generalization performance improves correspondingly. In Figure 4b, we analyze how the number of bandwidths $K$ affects the performance of UniOD. A larger $K$ results in less information loss, which improves the generalization ability. These results are consistent with Theorem 4.1. For more detailed results, please refer to Appendix H.

## 6 CONCLUSION

This work proposed a novel and efficient outlier detection method called UniOD. The core idea of UniOD is to leverage historical datasets to train a deep universal model that can detect outliers in newly unseen datasets from diverse domains without retraining. By converting each dataset into graph-structured data and generating uniformly dimensioned node features, UniOD enables a single model to handle heterogeneous datasets. We provide both theoretical analysis and empirical results to demonstrate its effectiveness and efficiency. Although UniOD is primarily designed for transductive anomaly detection, it can also be applied to inductive anomaly detection by converting the training set and each test point into graph-structured data and computing their outlier scores.

## ACKNOWLEDGMENTS

The work was partially supported by the National Natural Science Foundation of China under Grant No.62376236, the General Program of the Natural Science Foundation of Guangdong Province under Grant No.2024A1515011771, and the Shenzhen Stability Science Program 2023.

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

## A PROOF FOR THEOREM 4.1

Recall that our model $f \in \mathcal{F}$ is composed of $K$ GINs and $K$ GTs followed by an MLP. For convenience, in the main paper, we have let each GIN have $L$ layers and each layer has an MLP of depth $L'$. The GTs have the same sizes. The depth of the MLP following the concatenation of GINs and GTs is also $L'$. We have also let $b_A = \max_{i,k} \|\mathbf{A}_{H_i,\sigma_k}\|_2$, $c_X = \max_k \sqrt{\sum_i \|\mathbf{X}_{H_i,\sigma_k}\|_F^2}$, $b_W = \max_{\mathbf{W} \in \mathcal{W}} \|\mathbf{W}\|_2$, $b'_W = \max_{\mathbf{W} \in \mathcal{W}} \|\mathbf{W}\|_{2,1}$. Suppose all activation functions are 1-Lipschitz, and $\ell$ is $\mu$-Lipschitz and bounded by $\beta$.

The main idea of our analysis is composed of the following steps: 1) derive the covering number bound of $\mathcal{F}$; 2) derive the Rademacher complexity bound of the loss function; 3) establish the generalization bound of $f$.

First, we analyze the complexity of the input data $\mathbf{X}_k \in \mathbb{R}^{Mn \times d}$, which is composed of $\{\mathbf{X}_{H_i,\sigma_k}\}_{i=1}^M$, $k = 1, 2, \dots, K$. We show the following lemma (Lemma 3.2 in (Bartlett et al., 2017)).

**Lemma A.1.** *Let conjugate exponents $(p, q)$ and $(r, s)$ be given with $p \leq 2$, as well as positive reals $(a, b, \epsilon)$ and positive integer m. Let matrix $\mathbf{X} \in \mathbb{R}^{n \times d}$ be given with $\|\mathbf{X}\|_p \leq b$. Then*

$$\ln \mathcal{N}\left(\left\{\mathbf{XA} : \mathbf{A} \in \mathbb{R}^{d \times m}, \|\mathbf{A}\|_{q,s} \leq a\right\}, \epsilon, \|\cdot\|_F\right) \leq \left\lceil \frac{a^2 b^2 m^{2/r}}{\epsilon^2} \right\rceil \ln(2dm)$$

For convenience, we drop the index of $\mathbf{X}$. Denote $\mathcal{X} := \{\mathbf{X} : \|\mathbf{X}\|_F \leq c_X\}$. According to the lemma, letting $\mathbf{A}$ be an identity matrix (a virtual parameter matrix), and letting $p = q = 2, s = 1, r = \infty$, we have

$$\ln \mathcal{N}(\mathcal{X}, \epsilon, \|\cdot\|_F) \leq \frac{c_X^2 d^2}{\epsilon^2} \ln(2d^2) \tag{13}$$

The Lipschitz constant $\tau_{\text{GT}}$ of one GT is given by the following lemma (proved in Section B.1).

**Lemma A.2.** *Suppose in the GT, the layer normalization is $\rho$-Lipschitz. Let $\mathbf{Z}_{p-1}$ be the input of layer $p$, $\|\mathbf{Z}_{p-1}\|_2 \leq b_Z^{(p-1)}$, and $\|\mathbf{Z}_{p-1}\|_F \leq c_Z^{(p-1)}$. Then the Lipschitz constant of the GT is*

$$\tau_{GT} = \prod_{i=1}^{L} \left( u b_Z^{(i-1)} + v \right) \tag{14}$$

*where $u = \frac{\rho b_W^{3+L'} \sqrt{H}}{\sqrt{\bar{d}}}$ and $v = \rho b_W^{L'}(1 + \sqrt{H} b_W \sqrt{m})$ and $b_Z^{(i-1)} \leq c_Z^{(i-1)} \leq \tau_{GT_{i-1}} c_Z^{(i-2)}$. If all self-attentions are $\eta$-Lipschitz, then*

$$\tau_{GT} = b_W^{LL'} \rho_{norm}^L \left(1 + \sqrt{H}\eta\right)^L \tag{15}$$

According to Lemma A.3 (a well-known result), we can bound the covering numbers of the GT:

$$\ln \mathcal{N}(\mathcal{F}_{\text{GT}}, \epsilon, \|\cdot\|_F) \leq \frac{R_{\text{GT}}}{\epsilon^2} \tag{16}$$

where $R_{\text{GT}} = c_X^2 d^2 \tau_{\text{GT}}^2 \ln(2d^2)$.

**Lemma A.3.** *Suppose $\psi$ is a $\kappa$-Lipschitz function, then $\ln \mathcal{N}(\epsilon, \psi \circ \mathcal{F}, \rho) \leq \ln \mathcal{N}(\epsilon/\kappa, \mathcal{F}, \rho)$.*

The following lemma (proved in Section B.2) provides an upper bound for the covering number of a GIN model:

**Lemma A.4.** *Consider the L-layer GIN, where each layer has an MLP of $L'$ layers. For any $\epsilon > 0$, it has*

$$\ln \mathcal{N}(\epsilon, \mathcal{F}_G, \rho) \leq \frac{R_{GIN}}{\epsilon^2}$$

*where $R_{GIN} = b_A^{2L} c_X^2 b_W^{2LL'} L^3 L'^3 (b'_W/b_W)^2 \ln(2\bar{d}^2)$.*

Since our model is a concatenation of $K$ independent GINs and $K$ independent GTs, the covering number bound of their combination is

$$\ln \mathcal{N}(\mathcal{F}_K, \epsilon, \|\cdot\|_F) \leq \frac{K(R_{\text{GIN}} + R_{\text{GT}})}{\epsilon^2} \tag{17}$$

The K-GINs and K-GTs are followed by an MLP with $L'$ layers, and the Lipschitz constant of the MLP is $\tau_{\text{MLP}} = b_W^{L'}$. Consequently,

$$\ln \mathcal{N}\left(\mathcal{F}, \epsilon, \|\cdot\|_F\right) \leq \frac{K\tau_{\text{MLP}}^2(R_{\text{GIN}} + R_{\text{GT}})}{\epsilon^2} \tag{18}$$

Since the overall loss is $\hat{\mathcal{L}}(f) := \frac{1}{M}\sum_{i=1}^M \frac{1}{n}\sum_{j=1}^n \ell(\mathbf{y}_{H_i}^{(j)}, \hat{\mathbf{y}}_{H_i}^{(j)})$, the loss on one dataset is $\bar{e}(\mathbf{y}, \hat{\mathbf{y}}) = \frac{1}{n}\sum_{j=1}^n \ell(y_j, \hat{y}_j)$. Suppose $\ell$ is $\mu$-Lipschitz with respect to $y$. Then according to Lemma B.3, $\bar{e}$ is $\frac{\mu}{\sqrt{n}}$-Lipschitz. Then the covering number of our model with loss is bounded as

$$\ln \mathcal{N}\left(\ell \circ \mathcal{F}, \epsilon, \|\cdot\|_F\right) \leq \frac{K\mu^2\tau_{\text{MLP}}^2(R_{\text{GIN}} + R_{\text{GT}})}{n\epsilon^2} \tag{19}$$

The Dudley entropy integral bound used by Bartlett et al. (2017) is shown below.

**Lemma A.5** (Lemma A.5 of Bartlett et al. (2017)). *Let $\mathcal{F}$ be a real-valued function class taking values in $[0, 1]$, and assume that $\mathbf{0} \in \mathcal{F}$. Then*

$$\mathcal{R}_{\mathcal{D}}(\mathcal{F}) \leq \inf_{\alpha > 0}\left(\frac{4\alpha}{\sqrt{M}} + \frac{12}{M}\int_\alpha^{\sqrt{M}} \sqrt{\ln \mathcal{N}\left(\epsilon, \mathcal{F}, \rho\right)}\,d\epsilon\right).$$

In our method, $|\ell| \leq \beta$. We let $\tilde{\mathcal{F}} = \phi \circ \mathcal{F}$, where $\phi(x) = \frac{1}{\beta}x$, then $\tilde{\mathcal{F}}$ is a real-valued function class taking values in $[0, 1]$. Using Lemma A.5, we obtain

$$\begin{aligned}
\mathcal{R}_{\mathcal{D}}(\tilde{\mathcal{F}}) &\leq \inf_{\alpha > 0}\left(\frac{4\alpha}{\sqrt{M}} + \frac{12}{M}\int_\alpha^{\sqrt{M}} \sqrt{\ln \mathcal{N}\left(\frac{\epsilon}{\beta}, \tilde{\mathcal{F}}, \rho\right)}\,d(\frac{\epsilon}{\beta})\right) \\
&= \inf_{\alpha > 0}\left(\frac{4\alpha}{\sqrt{M}} + \frac{12}{M\beta}\int_{\beta\alpha}^{\beta\sqrt{M}} \sqrt{\ln \mathcal{M}\left(\frac{\epsilon}{\beta}, \phi \circ \mathcal{F}, \rho\right)}\,d\epsilon\right) \\
&\leq \inf_{\alpha > 0}\left(\frac{4\alpha}{\sqrt{M}} + \frac{12}{M\beta}\int_{\beta\alpha}^{\beta\sqrt{M}} \sqrt{\ln \mathcal{M}\left(\epsilon, \mathcal{F}, \rho\right)}\,d\epsilon\right)
\end{aligned} \tag{20}$$

Multiplying both side by $\beta$ yields

$$\mathcal{R}_{\mathcal{D}}(\mathcal{F}) \leq \inf_{\alpha > 0}\left(\frac{4\alpha\beta}{\sqrt{M}} + \frac{12}{M}\int_{\beta\alpha}^{\beta\sqrt{M}} \sqrt{\ln \mathcal{N}\left(\epsilon, \mathcal{F}, \rho\right)}\,d\epsilon\right). \tag{21}$$

Letting $v = \frac{K\mu^2\tau_{\text{MLP}}^2(R_{\text{GIN}} + R_{\text{GT}})}{n}$, it follows that

$$\begin{aligned}
\mathcal{R}_{\mathcal{G}}(\mathcal{F}) &\leq \inf_{\alpha > 0}\left(\frac{4\alpha\beta}{\sqrt{M}} + \frac{12}{M}\int_{\beta\alpha}^{\beta\sqrt{N}} \sqrt{\frac{v}{\epsilon^2}}\,d\epsilon\right) \\
&= \inf_{\alpha > 0}\left(\frac{4\alpha\beta}{\sqrt{M}} + \frac{12\sqrt{v}}{\ln}\left(\frac{\sqrt{M}}{\alpha}\right)\right) \\
&= \frac{4\beta}{M} + \frac{12\sqrt{v}\ln(M)}{M}
\end{aligned} \tag{22}$$

where we have chosen $\alpha = 1/\sqrt{M}$.

Then, using the following lemma (a standard tool in Rademacher complexity (Mohri et al., 2018)), the generalization bound of our UniOD can be derived.

**Lemma A.6.** *Given hypothesis function space $\mathcal{F}$ and loss function $\ell_\beta$ bounded by $\beta > 0$, define $\mathcal{F}_\beta := \{(\mathbf{Z}, \mathbf{y}) \mapsto \ell_\beta(f(\mathbf{Z}), \mathbf{y}) : f \in \mathcal{F}\}$. Then, with probability at least $1 - \delta$ over a sample $\mathcal{D}$ of size $M$, every $f \in \mathcal{F}$ satisfies $\mathcal{L}_\beta(f) \leq \hat{\mathcal{L}}_\beta(f) + 2\mathcal{R}_{\mathcal{D}}(\mathcal{F}_\beta) + 3\beta\sqrt{\frac{\ln(2/\delta)}{2M}}$.*

Specifically, we have

$$
\begin{aligned}
\mathcal{L}(f) \leq & \hat{\mathcal{L}}(f) + \frac{8\beta\sqrt{n} + 24\sqrt{K}\mu\tau_{\text{MLP}}\sqrt{R_{\text{GIN}} + R_{\text{GT}}}\ln(M)}{M\sqrt{n}} + 3\beta\sqrt{\frac{\ln(2/\delta)}{2M}} \\
\leq & \hat{\mathcal{L}}(f) + \frac{8\beta\sqrt{n} + 24\sqrt{K}\mu b_W^{L'}(\sqrt{R_{\text{GIN}}} + \sqrt{R_{\text{GT}}})\ln(M)}{M\sqrt{n}} + 3\beta\sqrt{\frac{\ln(2/\delta)}{2M}}
\end{aligned}
\tag{23}
$$

where $R_{\text{GIN}} = b_A^{2L}c_X^2 b_W^{2LL'}L^3 L'^3 (b_W'/b_W)^2\ln(2\bar{d}^2)$, $R_{\text{GT}} = c_X^2 d^2 \tau_{\text{GT}}^2 \ln(2d^2)$, $\tau_{\text{GT}} = \prod_{i=1}^{L}\left(ub_Z^{(i-1)} + v\right)$, $u = \frac{\rho b_W^{3+L'}\sqrt{H}}{\sqrt{\bar{d}}}$, $v = \rho b_W^{L'}(1 + \sqrt{H}b_W\sqrt{n})$, and $b_Z^{(i-1)} \leq c_Z^{(i-1)} \leq \tau_{\text{GT}_{i-1}}c_Z^{(i-2)} \leq \prod_{j=1}^{i-2}\left(ub_Z^{(j)} + v\right)c_X$. Dropping the terms related to the residual connections and layer normalizations in the GTs, we complete the proof.

If we assume that all self-attentions are $\eta$-Lipschitz, then $\tau_{\text{GT}} = b_W^{LL'}\rho_{norm}^L\left(1 + \sqrt{H}\eta\right)^L$.

# B LEMMAS AND PROOFS

## B.1 PROOF FOR LEMMA A.2

*Proof.* We denote $f_i$ the $i$-th layer of a GT. It is composed of multi-head self-attention, residual connection, layer normalization, and MLP. To analyze the covering number bound of a single GT, we first present the following lemma for the Lipschitz continuity of self-attention.

**Lemma B.1.** *Consider the following self-attention mechanism*

$$
f(\mathbf{Z}) = \textit{Softmax}\left(\frac{\mathbf{Z}\mathbf{W}_Q(\mathbf{Z}\mathbf{W}_K)^\top}{\sqrt{d}}\right)\mathbf{Z}\mathbf{W}_V
$$

*where $\mathbf{Z} \in \mathbb{R}^{n\times d'}$ and $\mathbf{W}_Q, \mathbf{W}_K, \mathbf{W}_V \in \mathbb{R}^{d'\times d}$. Suppose $\max\{\|\mathbf{W}_Q\|_2, \|\mathbf{W}_K\|_2, \|\mathbf{W}_V\|_2\} \leq b_{att}$, then for any $\mathbf{Z}$ and $\mathbf{Z}'$ satisfying $\|\mathbf{Z}\|_2 \leq b_z$ and $\|\mathbf{Z}'\|_2 \leq b_z$, it holds that*

$$
\|f(\mathbf{Z}) - f(\mathbf{Z}')\| \leq \left(\frac{1}{\sqrt{d}}b_{att}^3 b_z + b_{att}\sqrt{n}\right)\|\mathbf{Z} - \mathbf{Z}'\|_F
\tag{24}
$$

Then, according to the concatenation Lipschitz lemma (Lemma B.4), the Lipschitz constant of the multi-head ($H$ heads) attention is

$$
L_{\text{multi-att}} = \sqrt{H}\left(\frac{1}{\sqrt{d}}b_{att}^3 b_z + b_{att}\sqrt{n}\right)
\tag{25}
$$

Assume that the Lipschitz constant of the layer normalization is $\rho_{norm}$, and the feedforward network has $L'$ layers. Then the Lipschitz constant of the transformer layer $i$ is

$$
\begin{aligned}
L_{\text{GT}_i} = & b_W^{L'}\rho_{norm}\left(1 + \sqrt{H}\left(\frac{1}{\sqrt{d}}b_{att}^3 b_z^{(i-1)} + b_{att}\sqrt{n}\right)\right) \\
= & ub_z^{(i-1)} + v
\end{aligned}
\tag{26}
$$

where $u = \frac{b_W^{L'}b_{att}^3\rho_{norm}\sqrt{H}}{\sqrt{d}}$ and $v = b_W^{L'}\rho_{norm}(1 + \sqrt{H}b_{att}\sqrt{n})$. As $\|\mathbf{Z}_i\|_2 \leq b_z^{(i)}$ and $\|\mathbf{Z}_i\|_2 \leq \|\mathbf{Z}_i\|_F$, we bound $\|\mathbf{Z}_i\|_F$ instead, where $\|\mathbf{Z}_0\|_F = \|\mathbf{X}\|_F \leq c_X$. That means, we let $b_z^{i-1}$ be the upper bound of $\|\mathbf{Z}_{i-1}\|_F$. It follows that

$$
b_z^{i-1} \leq L_{\text{GT}_{i-1}}b_z^{i-2}
\tag{27}
$$

We have $L$ layers in the GIN. Therefore,

$$
L_{\text{GT}} = \prod_{i=1}^{L}(ub_z^{(i-1)} + v)
\tag{28}
$$

If all self-attentions are $\eta$-Lipschitz, i.e.,

$$\|f(\mathbf{Z}) - f(\mathbf{Z}')\| \leq \eta \|\mathbf{Z} - \mathbf{Z}'\|_F \tag{29}$$

then the Lipschitz constant of the transformer layer $i$ is

$$L_{\mathrm{GT}_i} = b_W^{L'} \rho_{norm} \left(1 + \sqrt{H}\eta\right) \tag{30}$$

which makes

$$L_{\mathrm{GT}} = b_W^{LL'} \rho_{norm}^L \left(1 + \sqrt{H}\eta\right)^L \tag{31}$$

$\square$

## B.2  PROOF FOR LEMMA A.4

Since the lemma is a simple variant of Lemma F.8 in (Wang & Fan, 2024), we will not detail the proof here. For convenience, we show the original lemma (with modified notations) below:

**Lemma B.2** (Covering number bound of GIN). *Let $c = \|\tilde{\mathbf{A}}\|_2$ and $\bar{d} = \max_{i,l} d_i^{(l)}$. Given an $L$-layer GIN message passing network $\mathcal{F}_G$, for any $\epsilon > 0$*

$$\ln \mathcal{N}\left(\epsilon, \mathcal{F}_G, \rho\right) \leq \frac{R_G}{\epsilon^2}$$

*where $R_G = c^{2L} \|\mathbf{X}\|_F^2 \ln(2\bar{d}^2) \left(\prod_{l=1}^{L} \kappa_l^2\right) \left(\sum_{l=1}^{L} (\tau_l)^{\frac{2}{3}}\right)^3$ and $\kappa_l = \prod_{j=1}^{r} \kappa_j^{(l)}$, $\tau_l = \left(\sum_{i=1}^{r} \left(\frac{b_i^{(l)}}{\kappa_i^{(l)}}\right)^{2/3}\right)^{3/2}$.*

In the lemma, let $r = L'$, $c = b_A$, $b_i^{(l)} = b_W'$, and $\kappa_i^{(l)} = b_W$, we obtain Lemma A.4.

## B.3  PROOF FOR LEMMA B.3

**Lemma B.3.** *Let $\bar{e}(\mathbf{y}) = [\frac{1}{n} \sum_{j=1}^{n} \ell(y_{ij}^*, y_{ij})]_i$. Suppose $\ell$ is $\mu$-Lipschitz, then $\bar{e}$ is $\frac{\mu}{\sqrt{n}}$-Lipschitz.*

*Proof.* It can be proved by the following derivations:

$$
\begin{aligned}
&\|\bar{e}(\mathbf{y}) - \bar{e}(\mathbf{y}')\| \\
=& \frac{1}{n} \left\| \begin{matrix} \sum_{j=1}^{n}(\ell(y_{1j}^*, y_{1j}) - \ell(y_{1j}^*, y_{1j}')) \\ \sum_{j=1}^{n}(\ell(y_{2j}^*, y_{2j}) - \ell(y_{2j}^*, y_{2j}')) \\ \cdots \\ \sum_{j=1}^{n}(\ell(y_{Nj}^*, y_{Nj}) - \ell(y_{Nj}^*, y_{Nj}')) \end{matrix} \right\| \\
\leq& \frac{1}{n} \left\| \begin{matrix} \sum_{j=1}^{n} \mu|y_{1j} - y_{1j}'| \\ \sum_{j=1}^{n} \mu|y_{2j} - y_{2j}'| \\ \cdots \\ \sum_{j=1}^{n} \mu|y_{Nj} - y_{Nj}'| \end{matrix} \right\| \\
\leq& \frac{\mu}{n} \left\| \begin{matrix} \sqrt{n}\|\mathbf{y}_1 - \mathbf{y}_1'\| \\ \sqrt{n}\|\mathbf{y}_2 - \mathbf{y}_2'\| \\ \cdots \\ \sqrt{n}\|\mathbf{y}_N - \mathbf{y}_N'\| \end{matrix} \right\| \\
=& \frac{\mu}{\sqrt{n}} \|\mathbf{y} - \mathbf{y}'\|
\end{aligned}
\tag{32}
$$

$\square$

### B.4 PROOF FOR LEMMA B.1

*Proof.* Let $g(\mathbf{Z}) = \frac{\mathbf{Z}\mathbf{W}^Q(\mathbf{Z}\mathbf{W}^K)^\top}{\sqrt{d}}$. We calculate

$$
\begin{aligned}
&\|f(\mathbf{Z}) - f(\mathbf{Z}')\|_F = \|\mathrm{Softmax}(g(\mathbf{Z}))\mathbf{Z}\mathbf{W}_V - \mathrm{Softmax}(g(\mathbf{Z}'))\mathbf{Z}'\mathbf{W}_V\|_F \\
&\leq \|\mathbf{W}_V\|_2 \|\mathrm{Softmax}(g(\mathbf{Z}))\mathbf{Z} - \mathrm{Softmax}(g(\mathbf{Z}'))\mathbf{Z}'\|_F \\
&\leq \|\mathbf{W}_V\|_2 \left(\|\mathrm{Softmax}(g(\mathbf{Z}))\mathbf{Z} - \mathrm{Softmax}(g(\mathbf{Z}'))\mathbf{Z}\|_F + \|\mathrm{Softmax}(g(\mathbf{Z}'))\mathbf{Z} - \mathrm{Softmax}(g(\mathbf{Z}'))\mathbf{Z}'\|_F\right) \\
&\leq \|\mathbf{W}_V\|_2 \left(\underbrace{\|\mathrm{Softmax}(g(\mathbf{Z})) - \mathrm{Softmax}(g(\mathbf{Z}'))\|_F}_{T_1} \|\mathbf{Z}\|_2 + \|\mathbf{Z} - \mathbf{Z}'\|_F \underbrace{\|\mathrm{Softmax}(g(\mathbf{Z}'))\|_2}_{T_2}\right)
\end{aligned}
\tag{33}
$$

For $T_1$, we have

$$
\begin{aligned}
T_1 &\leq \frac{1}{2}\|g(\mathbf{Z}) - g(\mathbf{Z}')\|_F \\
&= \frac{1}{2\sqrt{d}}\left\|\mathbf{Z}\mathbf{W}_Q(\mathbf{Z}\mathbf{W}_K)^\top - \mathbf{Z}'\mathbf{W}_Q(\mathbf{Z}'\mathbf{W}_K)^\top\right\|_F \\
&\leq \frac{1}{2\sqrt{d}}\left(\left\|\mathbf{Z}\mathbf{W}_Q\mathbf{W}_K^\top\mathbf{Z}^\top - \mathbf{Z}'\mathbf{W}_Q\mathbf{W}_K^\top\mathbf{Z}^\top\right\|_F + \left\|\mathbf{Z}'\mathbf{W}_Q\mathbf{W}_K^\top\mathbf{Z}^\top - \mathbf{Z}'\mathbf{W}_Q\mathbf{W}_K^\top\mathbf{Z}'^\top\right\|_F\right) \\
&\leq \frac{1}{2\sqrt{d}}\left(\|\mathbf{Z} - \mathbf{Z}'\|_F \|\mathbf{W}_Q\|_2 \|\mathbf{W}_K\|_2 \|\mathbf{Z}\|_2 + \|\mathbf{Z} - \mathbf{Z}'\|_F \|\mathbf{W}_Q\|_2 \|\mathbf{W}_K\|_2 \|\mathbf{Z}'\|_2\right) \\
&\leq \frac{1}{\sqrt{d}}b_{att}^2 b_z \|\mathbf{Z} - \mathbf{Z}'\|_F
\end{aligned}
\tag{34}
$$

For $T_2$, we have

$$
\begin{aligned}
T_2 &\leq \|\mathrm{Softmax}(g(\mathbf{Z}'))\|_F \\
&\leq \sqrt{\sum_{i=1}^m \|\mathrm{Softmax}(\mathbf{t}_i)\|^2} \\
&\leq \sqrt{\sum_{i=1}^m \|\mathrm{Softmax}(\mathbf{t}_i)\|_1^2} \\
&= \sqrt{m}
\end{aligned}
\tag{35}
$$

where $\mathbf{t}_i$ denotes $g_i(\mathbf{Z})$. Combining the results above, we obtain

$$
\|f(\mathbf{Z}) - f(\mathbf{Z}')\|_F \leq \left(\frac{1}{\sqrt{d}}b_{att}^3 b_z + b_{att}\sqrt{m}\right)\|\mathbf{Z} - \mathbf{Z}'\|_F
\tag{36}
$$

□

### B.5 PROOF FOR LEMMA B.4

**Lemma B.4** (Concatenation Lipschitz). *For $K$ matrix functions with the same input $\mathbf{X}$, suppose each of them has a Lipschitz constant $L_k$. Then the concatenation of these functions has a Lipschitz constant $\max_k L_k \sqrt{K}$ with respect to $\mathbf{X}$.*

*Proof.* Given $K$ matrix functions $f_1, f_2, \ldots, f_K$ with the same input $\mathbf{X}$, the row-wise concatenation of the output is denoted $\bar{\mathbf{Y}}$, given by $F(\mathbf{X}) := [f_1(\mathbf{X}); f_2(\mathbf{X}); \ldots; f_K(\mathbf{X})]$. Suppose the Lipschitz

constant of each $f_k$ is $L_k$, $k = 1, 2, \ldots, K$. We obtain

$$
\begin{aligned}
\|F(\mathbf{X}) - F(\mathbf{X}')\| &= \left\| \begin{matrix} f_1(\mathbf{X}) - f_1(\mathbf{X}') \\ f_2(\mathbf{X}) - f_2(\mathbf{X}') \\ \vdots \\ f_K(\mathbf{X}) - f_K(\mathbf{X}') \end{matrix} \right\| \\
&= \sqrt{\sum_{k=1}^{K} \|f_k(\mathbf{X}) - f_k(\mathbf{X}')\|_F^2} \\
&\leq \sqrt{\sum_{k=1}^{K} L_k^2 \|\mathbf{X} - \mathbf{X}'\|_F^2} \\
&\leq \max_k L_k \sqrt{\sum_{k=1}^{K} \|\mathbf{X} - \mathbf{X}'\|_F^2} \\
&= \max_k L_k \sqrt{K} \|\mathbf{X} - \mathbf{X}'\|_F
\end{aligned}
\tag{37}
$$

$\square$

## C  ALGORITHM DETAILS

### C.1  $\mathrm{KGIN}_{\theta_1}$ AND $\mathrm{KGT}_{\theta_2}$

$\mathrm{KGIN}_{\theta_1}$ and $\mathrm{KGT}_{\theta_2}$ are combinations of $K$ GINs and GTs respectively. For the $k$-th GIN $\mathrm{GIN}_{\theta_1^k}$ and $\mathrm{GT}_{\theta_2^k}$ parametered by $\theta_1^k, \theta_2^k$, their outputs follows:

$$
\begin{aligned}
\mathbf{Z}_{H_i}^{\mathrm{GIN}_k} &= \mathrm{GIN}_{\theta_1^k} \left( \tilde{\mathbf{X}}_{H_i}, \mathcal{A}_{H_i} \right), \quad i = 1, \ldots, M \\
\mathbf{Z}_{H_i}^{\mathrm{GT}_k} &= \mathrm{GT}_{\theta_2^k} \left( \tilde{\mathbf{X}}_{H_i} \right), \quad i = 1, \ldots, M
\end{aligned}
\tag{38}
$$

Then, we have $\mathbf{Z}_{H_i}^{\mathrm{GIN}}$ and $\mathbf{Z}_{H_i}^{\mathrm{GT}}$ follows:

$$
\begin{aligned}
\mathbf{Z}_{H_i}^{\mathrm{GIN}} &= [\mathbf{Z}_{H_i}^{\mathrm{GIN}_1}, \mathbf{Z}_{H_i}^{\mathrm{GIN}_2}, \ldots, \mathbf{Z}_{H_i}^{\mathrm{GIN}_K}], \theta_1 = \{\theta_1^k\}_{k=1}^K \\
\mathbf{Z}_{H_i}^{\mathrm{GT}} &= [\mathbf{Z}_{H_i}^{\mathrm{GT}_1}, \mathbf{Z}_{H_i}^{\mathrm{GT}_2}, \ldots, \mathbf{Z}_{H_i}^{\mathrm{GT}_K}], \theta_2 = \{\theta_2^k\}_{k=1}^K
\end{aligned}
\tag{39}
$$

### C.2  GIN

The details about GIN are as follows:

$$
\forall l \in [1, 2, \ldots, L-1], \ h^{(j),l+1} = f_{\Theta^{(l+1)}} \left( (1+\epsilon) h^{(j),l} + \sum_{u \in \mathcal{N}(j)} A h^{(u),l} \right)
\tag{40}
$$

where $h^{(j),l+1}$ is the output from $l+1$ GIN layer, $h^{(j),0} = \tilde{\mathbf{x}}_{H_i, \sigma_k}^{(j)}$ is the input node feature, $\mathcal{N}(j)$ denotes the neighbor set of node $j$, $f_{\Theta^{(l+1)}}$ is a multi layer perceptrons (MLP) parameterized by $\Theta^{(l+1)}$, $\epsilon_k$ is a learnable parameter.

### C.3  DETAILED TRAINING AND TESTING ALGORITHM

The detailed training and testing procedures of UniOD is provided in Algorithm 1.

## D  DETAILED COMPLEXITY COMPARISON FOR DEEP-LEARNING BASED OD METHODS

In this section, we analysis the time complexity on conducting outlier detection for each deep-learning based methods. For all these methods which primarily uses MLP in the form of encoder, we assume

---

**Algorithm 1** Training and Testing Stages of UniOD

---

**Training stage of UniOD:**

**Require:** historical datasets $\mathscr{D}_H = \{\mathcal{D}_{H_1}, \mathcal{D}_{H_2}, \ldots, \mathcal{D}_{H_M}\}$, training epoch $Q$, $\hat{\mathscr{X}}_H = \emptyset$, $\hat{\mathscr{A}}_H = \emptyset$

    **Output:**$\theta^*$

1: **for** each historical dataset $\mathcal{D}_{H_i} \in \mathscr{D}_H$ **do**
2:     $\{\mathcal{D}_{H_i^1}, \mathcal{D}_{H_i^2}, \mathcal{D}_{H_i^3}, \mathcal{D}_{H_i^4}, \mathcal{D}_{H_i^5}\} = Subsapmling(\mathcal{D}_{H_i})$
3:     Obtain $\hat{\mathscr{A}}_H = \hat{\mathscr{A}}_H \cup \{\mathcal{A}_{H_i^m}\}_{m=1}^5$, $\hat{\mathscr{X}}_H = \hat{\mathscr{X}}_H \cup \{\tilde{X}_{H_i^m}\}_{m=1}^5$ using (2) and (3) respectively.
4: **end for**
5: Initialize the parameters of UniOD: $\theta$
6: **for** $b = 1, \ldots, Q$ **do**
7:     **for** each $\mathbf{X}_{H_i^j} \in \hat{\mathscr{X}}_H$, $\mathcal{A}_{H_i^j} \in \hat{\mathscr{A}}_H$ **do**
8:         Obtain node classification prediction using (6, 7,8 and 9)
9:         Update parameters $\theta$ using (10)
10:     **end for**
11: **end for**

**Testing stage of UniOD:**

**Require:** newly unseen test dataset $\mathcal{D}_{T_i}$, trained model parameters $\theta^*$

    **Output:**$\{Outlier\ Score(\mathbf{x}_{T_i}^{(j)})\}_{j=1}^{n_{T_i}}$

12: Obatain graph structured data $\tilde{X}_{T_i}$, $\mathcal{A}_{T_i}$ using (2and 3)
13: Obtain outlier score $\{Outlier\ Score(\mathbf{x}_{T_i}^{(j)})\}_{j=1}^{n_{T_i}}$ 11 using (6, 7, 8 and 9)

---

that the largest hidden dimension is $\bar{d}$, the training epoch is $Q$ and the layer of this encoder is $L_4$, for those methods using decoder as well, we assume the layer of encoder is equal to $L_4$. The time complexity comparison of conducting outlier detection on a new dataset $\mathcal{D}_{T_i}$ between several deep-learning based methods is shown in Table 5. Note that here we assume Several specific parameters for these models are:

- SLAD: SLAD includes two hyperparamters $r, c$ which defines the repeat time for data trnasformation and the number of sub-vectors.

- ICL: ICL includes a hyperparamter $0 < q \le d_{T_i}$ which splits each data into $d_{T_i} - k + 1$ pairs.

- NeutralAD: NeutralAD includes a hyperparamter $e$ which determines the number of learnable transformations.

- DSVDD: DSVDD requires using AE to pretrain its encoder.

Table 5: Time complexity comparison of UniOD and classical deep learning based methods in detecting the outliers of a new dataset $\mathcal{D}_{T_i}$.

|  | Time Complexity |
|---|---|
| SLAD | $\mathcal{O}\left(QnrcL_4\bar{d}\right)$ |
| DPAD | $\mathcal{O}\left(Qn\bar{d}(L\bar{d} + n)\right)$ |
| ICL | $\mathcal{O}\left(QndL\bar{d}^2\right)$ |
| NeutralAD | $\mathcal{O}\left(QneL_4\bar{d}^2\right)$ |
| DSVDD | $\mathcal{O}\left(QnL_4\bar{d}^2\right)$ |
| AE | $\mathcal{O}\left(Qn_{T_i}L_4\bar{d}^2\right)$ |
| UniOD | $\mathcal{O}\left(n^2(d + K\bar{d}) + n\bar{d}^2\bar{L} + n^2\bar{d}\bar{L}'\right)$ |

## E  HYPERPARAMETER COMPARISON OF OD METHODS

Most deep-learning based OD methods have several hyperparameters which can significantly affect their performance. In Table 6, we compare the number of hyperparameters of different deep-learning based methods. Notably, classical deep-learning methods require careful tuning of these hyperparameters when applied to newly unseen datasets. In contrast, when applied to newly unseen datasets, UniOD involves no hyperparameters to be tuned.

Table 6: The number of hyperparameters in recent deep-learning based methods and UniOD when applied to newly unseen datasets.

| Methods | Hyperparameters | Number |
|---------|-----------------|--------|
| DTE-NP | $k, T$ | 2 |
| SLAD | $h, c, r, \delta, \gamma,$ | 5 |
| DPAD | $\gamma, \lambda, k$ | 3 |
| ICL | $k, u, \tau, r$ | 4 |
| NeuralAD | $\tau, K, m$ | 3 |
| UniOD | - | 0 |

## F  STATISTICS OF DATASETS

In our experiments, we train our model using 15 historical datasets and evaluate the performance of 16 methods on 15 widely used real-world datasets spanning multiple domains, including healthcare, audio, language processing, and finance, in a popular benchmark for anomaly detection Han et al. (2022). The statistics of these datasets are shown in Table 7. These datasets encompass a range of samples and features, from small to large, providing comprehensive metrics and evaluations for the methods.

## G  HYPERPARAMETER COMBINATION OF OD METHODS

In our experiments, we perform a grid search on the historical datasets to identify the best hyperparameter configuration for each traditional and deep OD method, and use these configurations in subsequent evaluations. MetaOD further leverages these hyperparameter settings in traditional methods to train a model that predicts the optimal method–hyperparameter pair for each testing dataset. The specific hyperparameter configurations for different methods are summarized in Table 8.

## H  DETAILED EXPERIMENTAL SETTINGS AND RESULTS

In this section, we provide more detailed experimental settings and results.

### H.1  INTRODUCTION OF KPCA+MLP AND MLP+TF

For KPCA+MLP, KPCA (Schölkopf et al., 1998) is first applied to obtain uniformly dimensioned data across historical datasets. These transformed datasets are then used to train MLPs as classifiers. The main difference from UniOD is that similarity matrices are not directly utilized. Inspired by meta-learning approaches (Iwata & Kumagai, 2020; 2023; Hollmann et al., 2025), we also introduce another baseline, MLP+TF. In this method, each feature is first projected into a higher dimension using an MLP, then each sample can be represented as a sequence whose length equals the number of features, and whose token dimension is the MLP output size. A transformer is then applied to extract embeddings from these sequences, followed by a sum readout function to obtain a representation for each sequence (sample), which is subsequently fed into another MLP for classification.

Table 7: Statistics of 30 real-world datasets in ADBench.

| Data | # Samples | # Features | # Outlier | % Outlier Ratio | Category | Historical |
|---|---|---|---|---|---|---|
| ALOI | 49534 | 27 | 1508 | 3.04 | Image | ✓ |
| campaign | 41188 | 62 | 4640 | 11.27 | Finance | ✓ |
| cardio | 1831 | 21 | 176 | 9.61 | Healthcare | ✓ |
| celeba | 202599 | 39 | 4547 | 2.24 | Image | ✓ |
| cover | 286048 | 10 | 2747 | 0.96 | Botany | ✓ |
| Wilt | 4819 | 5 | 257 | 5.33 | Botany | ✓ |
| http | 567498 | 3 | 2211 | 0.39 | Web | ✓ |
| letter | 1600 | 32 | 100 | 6.25 | Image | ✓ |
| magic.gamma | 19020 | 10 | 6688 | 35.16 | Physical | ✓ |
| mammography | 11183 | 6 | 260 | 2.32 | Healthcare | ✓ |
| shuttle | 49097 | 9 | 3511 | 7.15 | Astronautics | ✓ |
| skin | 245057 | 3 | 50859 | 20.75 | Image | ✓ |
| smtp | 95156 | 3 | 30 | 0.03 | Web | ✓ |
| speech | 3686 | 400 | 61 | 1.65 | Linguistics | ✓ |
| vowels | 1456 | 12 | 50 | 3.43 | Linguistics | ✓ |
| breastw | 683 | 9 | 239 | 34.99 | Healthcare | × |
| Cardiotocography | 2114 | 21 | 466 | 22.04 | Healthcare | × |
| fault | 1941 | 27 | 673 | 34.67 | Physical | × |
| InternetAds | 1966 | 1555 | 368 | 18.72 | Image | × |
| landsat | 6435 | 36 | 1333 | 20.71 | Astronautics | × |
| optdigits | 5216 | 64 | 150 | 2.88 | Image | × |
| PageBlocks | 5393 | 10 | 510 | 9.46 | Document | × |
| pendigits | 6870 | 16 | 156 | 2.27 | Image | × |
| Pima | 768 | 8 | 268 | 34.90 | Healthcare | × |
| satellite | 6435 | 36 | 2036 | 31.64 | Astronautics | × |
| SpamBase | 4207 | 57 | 1679 | 39.91 | Document | × |
| satimage-2 | 5803 | 36 | 71 | 1.22 | Astronautics | × |
| thyroid | 3772 | 6 | 93 | 2.47 | Healthcare | × |
| Waveform | 3443 | 21 | 100 | 2.90 | Physics | × |
| WDBC | 367 | 30 | 10 | 2.72 | Healthcare | × |

Table 8: Hyperparameter combinations of traditional and deep OD methods used for grid search and MetaOD.

| Method | Hyperparameter 1 | Hyperparameter 2 | Total Methods |
|---|---|---|---|
| LOF | n_neighbors: [1, 5, 10, 15, 20, 25, 50, 60, 70, 80, 90, 100] | distance: ['manhattan', 'euclidean', 'minkowski'] | 36 |
| kNN | n_neighbors: [1, 5, 10, 15, 20, 25, 50, 60, 70, 80, 90, 100] | method: ['largest', 'mean', 'median'] | 36 |
| OCSVM | nu (train error tol): [0.1, 0.2, 0.3, 0.4, 0.5, 0.6, 0.7, 0.8, 0.9] | kernel: ['linear', 'poly', 'rbf', 'sigmoid'] | 36 |
| KDE | gamma: [0.3,0.5,1,3,5] | distance: ['manhattan', 'euclidean', 'minkowski'] | 15 |
| IF | n_estimators: [10, 20, 30, 40, 50, 75, 100, 150, 200] | max_features: [0.1, 0.2, 0.3, 0.4, 0.5, 0.6, 0.7, 0.8, 0.9] | 81 |
| LODA | n_bins: [10, 20, 30, 40, 50, 75, 100, 150, 200] | n_random_cuts: [5, 10, 15, 20, 25, 30] | 54 |
| AE | epoch_num: [10,50,100] | batch_size: [256,512] | 6 |
| DSVDD | l2_regularizer: [1e-2,1e-1] | epoch_num: [10,50,100] | 6 |
| NeutralAD | n_trans: [5,10,15,20] | temp: [0.3,0.5,0.8] | 12 |
| ICL | max_negatives: [100,300,500,1000] | temperature: [0.3,0.5,0.8] | 12 |
| SLAD | n_slad_ensemble: [10,20,30,40] | hidden_dims: [32,64,128] | 12 |
| DPAD | gama: [0.01,0.1,0.5,1] | lambda: [0.01,0.1,1] | 12 |
| DTE-NP | K: [5,10,15,20] | T: [200,500,1000,2000] | 16 |

## H.2 IMPLEMENTATION DETAILS

All of our experiments in this paper are implemented using Pytorch (Paszke et al., 2017) on a system equipped with an NVIDIA Tesla A40 GPU and an AMD EPYC 7543 CPU. In UniOD, we set the dimension of unified feature $d = 256$, the number of $\sigma$ is determined as $K = 5$, with $\{\beta_k^2\}_{k=1}^5 = \{0.3, 0.5, 1, 3, 5\}$. Our $\text{KGIN}_{\theta_1}$ comprises $L_1 = 4$ layers with successive hidden-dimensionalities $[1024, 1024, 1024, 64]$. As for $\text{KGT}_{\theta_2}$, the dimension of the feedforward network model is set to $1024$ and the layer is set to $L_2 = 6$. Classifier $\text{MLP}_{\theta_3}$ is a 3-layer MLP with successive hidden-dimensionalities $[128, 64, 2]$. We use AdamW (Loshchilov & Hutter, 2017) as our optimizer with learning rate being $5 * 10^{-5}$ and weight decay being $10^{-6}$ to train UniOD for 50 epochs.

## H.3 Experimental Results with more OD methods

The experimental results on more OD methods with hyperparameter tuning including the cross validation experiments is shown in Table 9 and Table 10.

Table 9: Complete average AUROC (%) and AUPRC (%) of each method on 15 tabular datasets of ADBench. The best results are marked in **bold**.

| AUROC | KDE (1962) | LOF (2000) | kNN (2000) | OC-SVM (2001) | AE (2006) | IF (2008) | LODA (2016) | DSVDD (2018) | NeutralAD (2021) | ECOD (2022) | ICL (2022) | SLAD (2023) | DPAD (2024) | DTE-NP (2024) | KPCA +MLP | MetaOD (2021) | UniOD Ours |
|---|---|---|---|---|---|---|---|---|---|---|---|---|---|---|---|---|---|
| breastw | 98.43 | 45.20 | 98.47 | 99.01 | 95.12 | 98.76 | 95.41 | 82.88 | 84.92 | 99.14 | 82.57 | 81.35 | 86.29 | 97.89 | 88.16 | 97.71 | 99.10 |
| Cardiotocography | 50.27 | 55.66 | 51.91 | 68.38 | 55.55 | 67.97 | 77.64 | 71.36 | 38.63 | 78.53 | 40.82 | 32.60 | 30.62 | 49.23 | 53.30 | 60.51 | 51.20 |
| fault | 73.05 | 59.59 | 71.46 | 53.95 | 66.25 | 56.15 | 41.56 | 48.94 | 67.35 | 46.87 | 55.29 | 71.83 | 66.77 | 73.44 | 90.55 | 57.21 | 69.60 |
| InternetAds | 61.79 | 62.55 | 62.36 | 61.65 | 53.71 | 66.67 | 46.62 | 62.18 | 66.42 | 67.70 | 42.73 | 64.66 | 53.86 | 65.11 | 58.18 | 69.62 | 63.50 |
| landsat | 62.46 | 52.86 | 61.58 | 41.16 | 49.87 | 52.78 | 42.98 | 42.59 | 61.92 | 36.78 | 69.91 | 67.24 | 52.72 | 59.20 | 39.04 | 56.8 | 69.10 |
| optdigits | 32.32 | 39.03 | 39.91 | 51.78 | 45.11 | 68.72 | 52.42 | 40.74 | 69.38 | 60.45 | 54.18 | 55.29 | 50.63 | 36.38 | 56.63 | 87.22 | 73.80 |
| PageBlocks | 90.66 | 85.05 | 92.08 | 90.37 | 91.63 | 89.52 | 67.94 | 89.72 | 84.17 | 91.39 | 62.34 | 74.08 | 52.70 | 89.54 | 76.26 | 75.99 | 88.20 |
| pendigits | 89.05 | 48.11 | 90.23 | 93.61 | 79.46 | 96.11 | 91.98 | 87.87 | 77.51 | 92.74 | 36.64 | 60.11 | 63.73 | 77.08 | 64.61 | 72.11 | 77.50 |
| Pima | 72.28 | 66.99 | 72.53 | 67.45 | 63.86 | 67.06 | 62.38 | 63.21 | 61.19 | 59.44 | 51.15 | 47.75 | 61.46 | 71.54 | 63.17 | 70.89 | 72.30 |
| satellite | 76.03 | 56.97 | 73.13 | 65.43 | 68.52 | 69.80 | 73.65 | 57.25 | 62.32 | 58.30 | 59.07 | 68.21 | 64.76 | 68.91 | 55.30 | 65.20 | 86.90 |
| satimage-2 | 96.44 | 68.67 | 99.88 | 99.55 | 95.71 | 99.42 | 99.72 | 96.68 | 79.71 | 96.49 | 64.50 | 95.97 | 81.22 | 96.59 | 59.62 | 91.51 | 99.70 |
| SpamBase | 49.52 | 38.79 | 56.14 | 54.25 | 55.99 | 62.92 | 45.47 | 56.82 | 51.09 | 65.56 | 21.68 | 52.66 | 47.14 | 53.14 | 53.39 | 66.20 | 56.30 |
| thyroid | 95.83 | 90.84 | 96.49 | 95.60 | 96.27 | 98.19 | 94.62 | 91.35 | 60.25 | 97.71 | 42.61 | 85.12 | 73.53 | 96.37 | 82.68 | 95.01 | 94.10 |
| Waveform | 75.12 | 75.59 | 75.35 | 70.18 | 60.25 | 70.72 | 63.68 | 64.70 | 69.31 | 60.35 | 62.36 | 44.54 | 55.22 | 73.89 | 62.04 | 69.40 | 84.30 |
| WDBC | 95.01 | 99.10 | 98.43 | 98.38 | 84.37 | 98.46 | 89.19 | 97.00 | 30.17 | 97.06 | 86.33 | 70.20 | 93.63 | 97.97 | 89.19 | 96.3 | 98.40 |
| Average Value | 74.55 | 63.00 | 76.00 | 74.05 | 70.78 | 77.55 | 69.68 | 70.22 | 64.29 | 73.90 | 55.48 | 64.77 | 62.29 | 73.75 | 66.13 | 75.45 | 78.93 |
| **AUPRC** | | | | | | | | | | | | | | | | | |
| breastw | 95.58 | 31.64 | 95.64 | 97.78 | 87.90 | 97.18 | 89.39 | 83.10 | 61.29 | 98.39 | 71.74 | 69.59 | 79.45 | 93.03 | 47.85 | 93.54 | 96.90 |
| Cardiotocography | 27.54 | 28.08 | 33.64 | 41.32 | 30.86 | 41.77 | 45.36 | 43.89 | 20.17 | 50.54 | 18.55 | 21.78 | 16.69 | 31.00 | 29.47 | 36.96 | 35.30 |
| fault | 54.57 | 40.22 | 52.08 | 39.26 | 48.16 | 41.34 | 30.10 | 33.39 | 47.69 | 32.57 | 39.29 | 53.04 | 47.89 | 53.77 | 84.08 | 42.77 | 49.10 |
| InternetAds | 23.80 | 34.28 | 29.76 | 29.47 | 19.33 | 42.56 | 25.04 | 29.83 | 34.68 | 50.89 | 16.07 | 29.35 | 21.11 | 29.39 | 28.39 | 52.50 | 32.80 |
| landsat | 26.03 | 23.44 | 25.72 | 17.75 | 20.23 | 21.13 | 17.71 | 19.02 | 28.33 | 16.35 | 42.64 | 29.21 | 22.14 | 25.13 | 15.95 | 24.29 | 28.10 |
| optdigits | 1.97 | 2.24 | 2.19 | 2.71 | 2.43 | 4.57 | 2.79 | 2.35 | 5.13 | 3.37 | 2.91 | 2.93 | 2.95 | 2.09 | 3.09 | 19.74 | 5.00 |
| PageBlocks | 53.98 | 47.06 | 56.83 | 50.74 | 52.12 | 48.81 | 27.73 | 51.98 | 32.69 | 51.99 | 31.58 | 36.85 | 11.82 | 51.11 | 30.61 | 29.08 | 46.20 |
| pendigits | 12.11 | 3.85 | 13.66 | 23.82 | 6.70 | 30.95 | 22.59 | 14.47 | 5.14 | 26.56 | 1.63 | 2.75 | 5.68 | 8.05 | 2.89 | 7.02 | 6.00 |
| Pima | 53.49 | 47.27 | 52.64 | 49.68 | 45.43 | 49.70 | 45.36 | 46.13 | 41.15 | 46.42 | 35.63 | 34.64 | 45.46 | 52.07 | 46.80 | 57.18 | 52.90 |
| satellite | 60.35 | 39.47 | 59.28 | 66.50 | 54.40 | 65.93 | 69.74 | 51.63 | 40.74 | 52.61 | 43.85 | 48.81 | 47.16 | 54.72 | 41.56 | 50.22 | 79.20 |
| satimage-2 | 31.80 | 2.81 | 94.52 | 96.28 | 28.47 | 93.69 | 83.68 | 78.87 | 2.85 | 65.97 | 1.59 | 37.66 | 8.42 | 40.75 | 43.99 | 29.83 | 95.70 |
| SpamBase | 38.29 | 33.72 | 41.36 | 40.61 | 41.91 | 48.23 | 37.92 | 43.23 | 38.63 | 51.83 | 26.72 | 40.66 | 38.82 | 40.38 | 40.41 | 51.20 | 42.50 |
| thyroid | 28.60 | 14.86 | 40.22 | 30.98 | 35.72 | 59.76 | 21.08 | 26.02 | 4.22 | 46.78 | 2.16 | 27.23 | 14.58 | 32.36 | 50.68 | 49.56 | 44.30 |
| Waveform | 11.44 | 13.39 | 13.26 | 5.64 | 4.25 | 5.63 | 4.35 | 4.61 | 33.56 | 4.05 | 6.19 | 2.41 | 4.22 | 11.35 | 5.78 | 4.84 | 9.60 |
| WDBC | 25.49 | 69.67 | 53.81 | 50.48 | 16.87 | 62.57 | 13.83 | 53.35 | 2.02 | 50.53 | 9.73 | 8.61 | 54.07 | 47.17 | 48.51 | 31.99 | 57.80 |
| Average Value | 36.34 | 28.80 | 44.31 | 42.87 | 32.99 | 47.59 | 35.78 | 38.79 | 26.55 | 43.26 | 23.35 | 29.70 | 28.03 | 38.16 | 34.67 | 38.71 | 45.43 |

Table 10: Complete average AUROC (%) and AUPRC (%) of each method on the 15 historical datasets described above, where the 15 test datasets described above are used as historical datasets. The best results are highlighted in **bold**.

| AUROC | KDE (1962) | LOF (2000) | kNN (2000) | OC-SVM (2001) | AE (2006) | IF (2008) | LODA (2016) | DSVDD (2018) | NeutralAD (2021) | ECOD (2022) | ICL (2022) | SLAD (2023) | DPAD (2024) | DTE-NP (2024) | KPCA +MLP | MetaOD (2021) | UniOD Ours |
|---|---|---|---|---|---|---|---|---|---|---|---|---|---|---|---|---|---|
| ALOI | 53.35 | 55.27 | 53.57 | 55.21 | 55.00 | 53.23 | 54.83 | 54.85 | 51.06 | 51.60 | 52.11 | 52.73 | 49.36 | 56.72 | 52.86 | 48.38 | 54.39 |
| campaign | 74.19 | 70.82 | 74.21 | 73.58 | 71.54 | 73.08 | 58.60 | 67.28 | 69.06 | 76.24 | 71.41 | 70.22 | 49.79 | 74.28 | 46.46 | 76.21 | 73.23 |
| cardio | 83.90 | 83.60 | 90.99 | 94.30 | 89.81 | 91.25 | 88.48 | 88.94 | 47.68 | 93.50 | 26.99 | 47.82 | 73.23 | 74.34 | 66.37 | 56.44 | 93.77 |
| celeba | 74.70 | 56.81 | 79.71 | 79.05 | 73.67 | 69.71 | 59.43 | 67.28 | 54.49 | 75.25 | 64.90 | 66.13 | 51.73 | 74.60 | 58.34 | 69.73 | 81.21 |
| cover | 87.00 | 96.13 | 91.40 | 92.96 | 89.32 | 90.04 | 89.45 | 73.72 | 89.64 | 54.04 | 66.15 | 69.82 | 91.63 | 62.69 | 88.82 | 93.52 |  |
| wilt | 34.02 | 54.94 | 44.40 | 32.30 | 44.98 | 44.91 | 40.85 | 31.93 | 52.72 | 39.40 | 57.57 | 59.83 | 51.08 | 58.00 | 57.32 | 51.72 | 52.61 |
| http | 99.57 | 99.47 | 99.41 | 99.87 | 99.79 | 99.46 | 99.70 | 99.80 | 98.27 | 35.05 | 99.95 | 51.61 | 12.54 | 99.90 | 99.95 | 100.00 |  |
| magic_gamma | 70.00 | 78.38 | 76.39 | 69.62 | 72.80 | 72.99 | 71.17 | 67.59 | 65.63 | 64.78 | 65.82 | 63.91 | 55.39 | 80.60 | 73.48 | 70.99 | 70.10 |
| mammography | 87.17 | 81.16 | 84.92 | 87.30 | 76.52 | 86.36 | 88.93 | 84.92 | 62.21 | 89.68 | 56.47 | 59.06 | 61.99 | 84.60 | 72.72 | 84.77 | 87.25 |
| shuttle | 99.37 | 44.50 | 92.06 | 99.24 | 99.05 | 99.83 | 87.65 | 99.26 | 73.58 | 99.40 | 52.21 | 94.27 | 56.66 | 79.58 | 80.61 | 49.17 | 99.51 |
| skin | 51.25 | 35.29 | 76.04 | 63.66 | 49.55 | 66.62 | 50.37 | 51.94 | 78.65 | 48.86 | 33.57 | 83.12 | 64.64 | 71.43 | 57.39 | 65.19 | 76.12 |
| speech | 52.02 | 47.86 | 47.83 | 46.87 | 47.49 | 46.67 | 44.17 | 45.70 | 53.61 | 46.97 | 43.22 | 52.41 | 50.98 | 49.99 | 48.30 | 54.88 | 46.97 |
| smtp | 100.00 | 99.99 | 100.00 | 100.00 | 99.98 | 98.89 | 83.34 | 100.00 | 99.69 | 100.00 | 99.94 | 100.00 | 74.25 | 100.00 | 99.99 | 97.81 | 100.00 |
| letter | 87.60 | 80.89 | 74.44 | 59.94 | 63.10 | 62.92 | 57.17 | 57.46 | 88.26 | 57.23 | 73.05 | 88.40 | 58.20 | 88.16 | 55.51 | 90.09 | 64.05 |
| vowels | 82.91 | 93.14 | 91.23 | 76.91 | 76.25 | 77.01 | 76.99 | 41.54 | 97.64 | 59.29 | 82.17 | 93.65 | 72.66 | 97.37 | 70.29 | 94.99 | 85.01 |
| Average Value | 75.80 | 71.88 | 78.45 | 75.36 | 73.93 | 75.55 | 70.31 | 70.50 | 70.99 | 72.67 | 57.90 | 73.18 | 59.43 | 72.92 | 67.24 | 73.28 | 78.52 |
| **AUPRC** | | | | | | | | | | | | | | | | | |
| ALOI | 4.26 | 4.32 | 3.97 | 4.28 | 4.26 | 3.60 | 4.31 | 3.96 | 4.38 | 3.30 | 3.69 | 3.79 | 3.03 | 4.76 | 3.41 | 2.79 | 3.62 |
| campaign | 28.03 | 22.63 | 28.49 | 28.27 | 25.03 | 32.06 | 16.03 | 24.55 | 20.93 | 34.27 | 22.88 | 23.95 | 11.35 | 28.82 | 10.89 | 36.70 | 28.47 |
| cardio | 36.47 | 28.77 | 51.18 | 57.10 | 43.13 | 52.60 | 47.62 | 46.35 | 11.93 | 56.74 | 6.14 | 19.69 | 30.85 | 35.49 | 23.86 | 18.95 | 57.63 |
| celeba | 5.86 | 2.56 | 9.27 | 11.37 | 5.59 | 5.91 | 3.20 | 6.24 | 2.31 | 10.76 | 4.99 | 5.29 | 2.37 | 6.34 | 3.63 | 5.37 | 11.71 |
| cover | 5.17 | 16.37 | 7.77 | 7.66 | 11.16 | 7.25 | 25.08 | 15.60 | 3.85 | 11.97 | 1.04 | 1.46 | 3.07 | 9.59 | 3.94 | 9.64 | 5.89 |
| wilt | 3.67 | 5.36 | 4.34 | 3.57 | 4.42 | 4.36 | 4.13 | 3.61 | 13.01 | 4.17 | 6.05 | 6.29 | 5.93 | 5.73 | 6.01 | 4.98 | 3.67 |
| http | 44.32 | 38.61 | 44.32 | 35.22 | 70.08 | 69.57 | 7.58 | 59.04 | 3.88 | 15.95 | 0.61 | 86.79 | 1.80 | 1.13 | 79.31 | 96.56 | 100.00 |
| magic_gamma | 63.52 | 65.71 | 69.83 | 63.70 | 64.36 | 65.22 | 62.34 | 56.21 | 50.11 | 54.50 | 55.26 | 52.45 | 40.51 | 73.46 | 64.54 | 62.54 | 62.48 |
| mammography | 20.98 | 12.99 | 15.98 | 17.22 | 12.14 | 24.44 | 28.86 | 21.19 | 3.05 | 41.16 | 3.95 | 4.10 | 6.07 | 16.69 | 26.42 | 21.87 | 19.50 |
| shuttle | 91.86 | 9.50 | 34.51 | 90.87 | 79.74 | 98.90 | 34.45 | 91.32 | 14.77 | 91.10 | 8.29 | 39.86 | 12.99 | 19.76 | 16.82 | 8.88 | 96.18 |
| skin | 19.08 | 15.00 | 31.60 | 23.78 | 19.47 | 25.17 | 18.91 | 18.24 | 43.32 | 18.24 | 14.92 | 46.87 | 24.57 | 28.48 | 23.99 | 24.53 | 31.65 |
| speech | 1.77 | 2.02 | 1.88 | 1.84 | 1.84 | 1.87 | 1.45 | 1.99 | 2.28 | 1.96 | 1.39 | 2.48 | 2.08 | 2.01 | 1.47 | 3.87 | 1.84 |
| smtp | 100.00 | 83.33 | 100.00 | 58.33 | 3.49 | 50.05 | 100.00 | 27.63 | 100.00 | 61.11 | 100.00 | 14.09 | 100.00 | 55.31 | 100.00 |  |  |
| letter | 35.42 | 24.27 | 15.30 | 10.09 | 11.30 | 8.96 | 8.66 | 9.44 | 42.13 | 7.67 | 17.96 | 36.87 | 8.99 | 30.41 | 6.64 | 52.40 | 10.54 |
| vowels | 23.17 | 38.42 | 30.29 | 16.33 | 21.48 | 14.43 | 15.70 | 4.02 | 56.95 | 8.14 | 27.68 | 33.96 | 18.11 | 56.06 | 5.81 | 35.61 | 17.23 |
| Average Value | 32.24 | 24.66 | 29.92 | 31.42 | 28.82 | 27.85 | 21.89 | 31.20 | 19.41 | 30.66 | 15.73 | 30.92 | 12.39 | 27.92 | 23.99 | 25.72 | 36.69 |

## H.4 Experimental Results with OD methods using their default hyperparameter combinations

In this subsection, we evaluate the effectiveness of our hyperparameter selection strategy by comparing the performance of OD methods under their default settings versus our selected hyperparameters. The results with default hyperparameters are shown in Figure 11. Several methods, including kNN, LOF,

OC-SVM, DSVDD, and NeutralAD, achieve better performance under our simple tuning strategy, while others exhibit slight performance degradation. This outcome is reasonable for two reasons: (i) the default or recommended hyperparameters in the original papers are already designed to be effective in most scenarios; and (ii) the randomly selected historical datasets may differ substantially from the test datasets in domain, feature semantics, and dimensionality, making it difficult to transfer well-tuned hyperparameters, such optimal hyperparamter different in different datasets can also be observed from Figure 2. This also explains that the performance of MetaOD is not as good as the most effective OD methods. Nevertheless, the fact that UniOD outperforms both default and tuned baselines highlights its robustness, even when historical datasets are not closely related to the test datasets.

Table 11: Average AUROC (%) and AUPRC (%) of each method using their default hyperparameter combinations on 15 tabular datasets of ADBench. The best results are marked in **bold**.

| AUROC | KDE (1962) | LOF (2000) | kNN (2000) | OC-SVM (2001) | AE (2006) | IF (2008) | DSVDD (2018) | NeutralAD (2021) | ECOD (2022) | ICL (2022) | SLAD (2023) | DPAD (2024) | DTE-NP (2024) | KPCA +MLP | UniOD Ours |
|---|---|---|---|---|---|---|---|---|---|---|---|---|---|---|---|
| breastw | 98.40 | 44.90 | 97.70 | 95.10 | 96.50 | 98.30 | 85.70 | 78.10 | 99.10 | 76.40 | 88.60 | 91.30 | 97.60 | 28.70 | **99.10** |
| Cardiotocography fault | 50.30 | 52.40 | 49.10 | 69.60 | 54.20 | 68.10 | 68.70 | 42.30 | 78.50 | 37.10 | 38.70 | 47.00 | 49.30 | 42.20 | 51.20 |
| InternetAds | 59.50 | 60.90 | 65.20 | 61.60 | 57.90 | 62.50 | 62.40 | 65.40 | 67.70 | 59.20 | 61.90 | 56.50 | 63.40 | 51.80 | 63.50 |
| landsat | 62.50 | 54.70 | 57.60 | 42.40 | 52.60 | 46.20 | 39.00 | **70.90** | 36.80 | 64.30 | 67.50 | 55.70 | 60.20 | 49.50 | 69.10 |
| opidigits | 32.30 | 53.70 | 37.20 | 50.70 | 47.20 | 69.60 | 33.80 | 34.70 | 60.50 | 53.30 | 54.80 | 46.50 | 38.60 | 39.80 | **73.80** |
| PageBlocks | 90.70 | 71.60 | 83.40 | 91.50 | 88.20 | 88.20 | 90.30 | 77.70 | **91.40** | 74.20 | 75.50 | 84.60 | 90.60 | 66.80 | 88.20 |
| pendigits | 72.30 | 89.10 | 94.70 | 74.30 | 93.10 | 49.90 | 77.50 | 70.40 | 92.70 | 67.30 | 66.60 | 64.90 | 78.60 | 62.40 | 72.30 |
| Pima | 72.30 | 60.10 | 70.90 | 62.40 | 62.80 | 67.40 | 65.30 | 57.10 | 59.40 | 51.50 | 51.30 | 65.40 | 70.70 | 53.70 | **72.30** |
| satellite | 76.00 | 54.20 | 66.50 | 66.40 | 66.70 | 50.50 | 62.80 | 57.60 | 58.30 | 60.10 | 73.40 | 68.50 | 70.20 | 49.90 | **86.90** |
| satimage-2 | 96.40 | 53.60 | 93.20 | 99.70 | 95.20 | 99.30 | 95.60 | 70.20 | 96.50 | 86.50 | 94.00 | 77.50 | 98.00 | 61.10 | **99.70** |
| SpamBase | 49.50 | 45.70 | 48.90 | 53.40 | 54.60 | **63.70** | 53.20 | 45.30 | 65.60 | 47.10 | 48.20 | 47.40 | 54.50 | 9.80 | 56.30 |
| thyroid | 95.80 | 66.50 | 95.90 | 95.90 | 94.10 | **97.90** | 91.40 | 64.50 | 97.70 | 73.20 | 80.90 | 84.50 | 96.40 | 73.00 | 94.10 |
| Waveform | 75.10 | 70.60 | 72.30 | 67.20 | 62.40 | 70.70 | 63.60 | 72.10 | 60.30 | 63.20 | 44.80 | 64.80 | 72.90 | 51.30 | **84.30** |
| WDBC | 95.00 | 98.20 | 97.40 | 98.80 | 94.40 | 98.80 | 97.70 | 26.90 | 97.10 | 75.10 | 87.60 | 83.70 | 97.50 | 58.50 | **99.10** |
| Average Value | 74.40 | 59.77 | 72.11 | 73.41 | 71.53 | 76.62 | 68.73 | 60.03 | 73.90 | 63.70 | 66.87 | 66.97 | 74.07 | 53.80 | **78.93** |
| **AUPRC** | | | | | | | | | | | | | | | |
| breastw | 95.50 | 29.70 | 92.30 | 91.70 | 90.50 | 96.20 | 84.60 | 53.50 | 98.30 | 75.70 | 78.40 | 77.70 | 92.10 | 26.90 | **96.90** |
| Cardiotocography fault | 27.50 | 27.50 | 28.60 | 41.30 | 30.00 | 43.50 | 42.70 | 21.20 | **50.50** | 16.60 | 23.60 | 25.00 | 31.20 | 22.60 | 35.30 |
| InternetAds | **54.50** | 39.60 | 52.90 | 40.10 | 51.50 | 39.70 | 36.90 | 49.20 | 32.50 | 45.30 | 51.90 | 48.40 | 53.20 | 31.40 | 49.10 |
| landsat | 22.60 | 24.20 | 27.90 | 29.20 | 22.00 | 47.00 | 30.20 | 27.80 | 50.80 | 25.10 | 26.90 | 24.20 | 29.00 | 20.00 | 32.80 |
| opidigits | 26.00 | 25.10 | 24.70 | 17.50 | 22.20 | 19.20 | 18.70 | 34.60 | 16.30 | **46.30** | 30.30 | 23.10 | 25.50 | 18.80 | 28.10 |
| PageBlocks | **5.00** | 3.50 | 2.20 | 2.60 | 2.50 | 4.70 | 2.00 | 2.00 | 3.30 | 3.02 | 29.00 | 2.60 | 2.10 | 20.10 | 1.90 |
| pendigits | 53.90 | 29.40 | 46.90 | 53.30 | 38.80 | 47.00 | **54.40** | 26.50 | 51.90 | 29.10 | 30.00 | 47.10 | 53.00 | 20.10 | 46.20 |
| Pima | 6.00 | 4.30 | 7.50 | 22.70 | 5.50 | 26.00 | 9.20 | 38.60 | 46.40 | 36.40 | 36.50 | 48.10 | 52.80 | 37.40 | **79.20** |
| satellite | 60.30 | 53.40 | 51.80 | 65.50 | 55.60 | 54.90 | 58.10 | 40.30 | 52.60 | 46.80 | 51.00 | 50.40 | 56.30 | 29.30 | **95.70** |
| satimage-2 | 31.80 | 3.10 | 34.70 | 96.50 | 32.20 | 91.60 | 56.30 | 2.19 | 65.90 | 10.30 | 27.80 | 4.50 | 50.70 | 1.40 | **95.70** |
| SpamBase | 38.20 | 35.90 | 39.40 | 40.20 | 40.90 | 48.70 | 39.70 | 37.40 | 51.80 | 31.30 | 37.60 | 38.20 | 40.70 | 49.80 | 42.50 |
| thyroid | 28.60 | 7.30 | 32.20 | 31.80 | 41.10 | 49.30 | 20.90 | 3.40 | 46.70 | 5.70 | 17.70 | 14.90 | 36.00 | 13.10 | **44.30** |
| Waveform | 9.60 | 11.40 | 4.70 | 10.50 | 5.30 | 7.50 | 5.00 | **27.20** | 4.00 | 7.60 | 2.40 | 6.00 | 10.90 | 2.70 | 9.60 |
| WDBC | 57.80 | 47.80 | 41.60 | 49.30 | 30.70 | 55.30 | 53.40 | 1.90 | 50.50 | 5.70 | 23.50 | 18.10 | 46.50 | 3.40 | **57.80** |
| Average Value | 36.26 | 24.30 | 36.12 | 42.25 | 34.15 | 45.89 | 37.50 | 24.71 | 43.26 | 25.88 | 30.33 | 28.93 | 39.26 | 18.79 | **45.43** |

## H.5 RESULTS OF USING MULTIPLY BAND WIDTH FOR SIMILARITY MATRICES CONSTRUCTION

Table 12 provides the OD performance results of UniOD on 15 tabular datasets of ADBench using different numbers $K$ of bandwidth for similarity matrices construction, the detection performance and generalization ability increase significantly, which mainly stems from less information loss of these datasets.

## H.6 RESULTS OF USING DIFFERENT NUMBERS OF HISTORICAL DATASETS

Table 13 presents the outlier-detection (OD) performance of UniOD on 15 tabular datasets from ADBench when it is trained with varying numbers of historical datasets, denoted by $M$. Even with $M = 1$, UniOD already achieves competitive performance on the 'breastw' dataset. This early success is largely attributable to the structural resemblance between the single historical dataset and 'breastw' after both of them are converted into graphs. As $M$ increases, UniOD is exposed to a more diverse set of graph-structured training examples, which enables the model to learn richer, structure-invariant representations of normal and anomalous patterns, thereby improving both its detection performance and its generalization ability to previously unseen datasets.

## H.7 RESULTS OF USING ORIGINAL HISTORICAL DATASETS

In the training of UniOD, we use a subsampling strategy on each historical dataset to create more data for training, which enhances the generalization capability of UniOD. In this subsection, we

Table 12: Complete average AUROC (%) and AUPRC (%) of UniOD on 15 tabular datasets of ADBench using different numbers $K$ of bandwidth for similarity matrices construction.

| AUROC | $K = 1$ | $K = 2$ | $K = 3$ | $K = 4$ | $K = 5$ |
|---|---|---|---|---|---|
| breastw | 96.73 | 96.83 | 77.35 | 98.36 | 99.10 |
| Cardiotocography | 44.49 | 41.76 | 49.69 | 47.21 | 51.20 |
| fault | 55.73 | 71.06 | 66.86 | 66.15 | 69.60 |
| InternetAds | 60.50 | 60.56 | 62.84 | 62.19 | 63.50 |
| landsat | 66.20 | 61.47 | 63.53 | 63.93 | 69.10 |
| optdigits | 81.61 | 71.68 | 60.58 | 58.88 | 73.80 |
| PageBlocks | 93.46 | 94.55 | 85.88 | 87.48 | 88.20 |
| pendigits | 53.94 | 78.17 | 88.41 | 85.06 | 77.50 |
| Pima | 70.72 | 74.34 | 73.66 | 72.80 | 72.30 |
| satellite | 84.59 | 81.13 | 82.48 | 82.62 | 86.90 |
| satimage-2 | 64.63 | 78.97 | 99.81 | 99.76 | 99.70 |
| SpamBase | 56.51 | 57.05 | 55.29 | 56.01 | 56.30 |
| thyroid | 97.69 | 96.23 | 90.70 | 96.94 | 94.10 |
| Waveform | 79.30 | 81.85 | 79.57 | 80.32 | 84.30 |
| WDBC | 87.20 | 74.90 | 95.29 | 97.17 | 98.40 |
| AVG | 72.89 | 74.70 | 75.46 | 76.99 | 78.93 |
| AUPRC | | | | | |
| breastw | 92.75 | 90.20 | 93.02 | 97.69 | 96.90 |
| Cardiotocography | 26.90 | 26.84 | 28.54 | 36.30 | 35.30 |
| fault | 42.14 | 53.49 | 52.78 | 45.45 | 49.10 |
| InternetAds | 29.07 | 30.52 | 30.36 | 29.94 | 32.80 |
| landsat | 26.90 | 24.84 | 29.80 | 20.84 | 28.10 |
| optdigits | 6.81 | 4.57 | 4.04 | 3.10 | 5.00 |
| PageBlocks | 57.53 | 73.25 | 45.73 | 43.95 | 46.20 |
| pendigits | 3.34 | 4.54 | 4.89 | 7.85 | 6.00 |
| Pima | 51.02 | 54.62 | 54.75 | 52.42 | 52.90 |
| satellite | 78.03 | 76.59 | 81.47 | 70.82 | 79.20 |
| satimage-2 | 7.97 | 19.50 | 49.60 | 96.24 | 95.70 |
| SpamBase | 42.92 | 44.22 | 42.26 | 41.07 | 42.50 |
| thyroid | 55.81 | 49.82 | 42.19 | 37.69 | 44.30 |
| Waveform | 8.40 | 16.87 | 14.94 | 6.96 | 9.60 |
| WDBC | 15.52 | 39.68 | 41.25 | 55.11 | 57.80 |
| AVG | 36.34 | 40.64 | 41.04 | 43.03 | 45.43 |

investigate how this influences the performance of UniOD by training UniOD using only the original historical datasets in Table 14.

## H.8    T-SNE PLOTS OF THE LEARNED REPRESENTATIONS $\mathbf{Z}_{T_i}$ ON MULTIPLE DATASETS.

In Figure 5, we provide t-SNE plots of the learned representations $\mathbf{Z}_{T_i}$ on multiple datasets to illustrate their structure. We observe that most outliers tend to concentrate into a small, dense cluster, while a smaller portion of outliers appear as isolated points.

We also notice that the separation between normal data and outliers is not particularly pronounced in the t-SNE space, which is likely because $\mathbf{Z}_{T_i}$ are high-dimensional representations (with dimensionality $> 1000$). As such, although a simple MLP can effectively separate normal and anomalous samples in this high-dimensional space, t-SNE may not faithfully preserve the underlying geometry of the original embedding space.

Table 13: Average AUROC (%) and AUPRC (%) of UniOD on 15 tabular datasets of ADBench using different numbers $M$ of historical datasets for training.

| AUROC | $M = 1$ | $M = 3$ | $M = 5$ | $M = 10$ | $M = 15$ |
|---|---|---|---|---|---|
| breastw | 84.59 | 15.39 | 70.53 | 98.39 | 99.10 |
| Cardiotocography | 28.37 | 64.32 | 57.71 | 61.71 | 51.20 |
| fault | 51.99 | 48.67 | 67.90 | 56.17 | 69.60 |
| InternetAds | 39.74 | 61.57 | 62.53 | 61.83 | 63.50 |
| landsat | 43.63 | 55.51 | 66.89 | 60.28 | 69.10 |
| optdigits | 43.15 | 70.63 | 66.30 | 62.65 | 73.80 |
| PageBlocks | 20.66 | 88.03 | 76.74 | 89.66 | 88.20 |
| pendigits | 7.19 | 56.70 | 79.32 | 87.98 | 77.50 |
| Pima | 37.73 | 67.64 | 72.58 | 71.18 | 72.30 |
| satellite | 28.42 | 76.14 | 84.95 | 80.18 | 86.90 |
| satimage-2 | 7.10 | 98.12 | 99.65 | 99.84 | 99.70 |
| SpamBase | 53.85 | 56.11 | 55.42 | 55.60 | 56.30 |
| thyroid | 14.78 | 85.55 | 95.82 | 96.66 | 94.10 |
| Waveform | 31.75 | 61.11 | 77.37 | 81.22 | 84.30 |
| WDBC | 14.79 | 95.10 | 98.60 | 96.67 | 98.40 |
| AVG | 33.85 | 66.71 | 75.49 | 77.33 | 78.93 |
| AUPRC | | | | | |
| breastw | 60.13 | 33.80 | 75.76 | 95.80 | 96.90 |
| Cardiotocography | 14.70 | 39.23 | 37.57 | 38.11 | 35.30 |
| fault | 34.20 | 31.96 | 48.42 | 42.63 | 49.10 |
| InternetAds | 16.35 | 33.11 | 30.88 | 29.74 | 32.80 |
| landsat | 17.39 | 23.01 | 27.22 | 24.05 | 28.10 |
| optdigits | 2.31 | 5.07 | 3.82 | 3.46 | 5.00 |
| PageBlocks | 5.57 | 45.37 | 24.62 | 45.44 | 46.20 |
| pendigits | 1.21 | 4.25 | 5.05 | 8.45 | 6.00 |
| Pima | 30.47 | 49.20 | 53.20 | 53.63 | 52.90 |
| satellite | 22.04 | 74.01 | 79.25 | 75.38 | 79.20 |
| satimage-2 | 0.65 | 58.84 | 92.87 | 96.69 | 95.70 |
| SpamBase | 48.92 | 42.50 | 41.31 | 41.05 | 42.50 |
| thyroid | 1.37 | 34.75 | 35.75 | 35.44 | 44.30 |
| Waveform | 1.94 | 3.55 | 6.22 | 9.00 | 9.60 |
| WDBC | 1.72 | 25.55 | 59.87 | 33.19 | 57.80 |
| AVG | 17.26 | 33.61 | 41.45 | 42.14 | 45.43 |

## H.9    ROBUSTNESS EVALUATION OF UNIOD TO THE DOMAIN OF HISTORICAL DATASETS

In this subsection, we conducted additional experiments where UniOD is evaluated on datasets from the physical, astronautics, and image domains, while systematically removing all historical datasets belonging to the same domain or field during training. As shown in Table 15, we observe that excluding these domain-specific datasets does not lead to a significant performance drop on the corresponding test domain, suggesting that UniOD is not overly sensitive to the particular composition of the historical training data.

We attribute this robustness to two main factors: (i) Even among tabular datasets within the same domain, the feature spaces and data characteristics can vary substantially. (ii) UniOD does not directly rely on the original raw features. Instead, it leverages similarity matrices to construct uniformly dimensioned representations across datasets. As a result, datasets from different domains may still exhibit similar structural patterns in their similarity matrices, enabling effective cross-domain generalization. Therefore, for a domain without historical data, our model, learned from the historical data of other domains, performs well.

Table 14: Average AUROC (%) and AUPRC (%) of UniOD on 15 tabular datasets of ADBench using only original historical datasets for training.

| AUROC | Original | Subsampling |
|---|---|---|
| breastw | 98.89 | 99.10 |
| Cardiotocography | 69.05 | 51.20 |
| fault | 51.28 | 69.60 |
| InternetAds | 61.62 | 63.50 |
| landsat | 47.11 | 69.10 |
| optdigits | 52.45 | 73.80 |
| PageBlocks | 89.94 | 88.20 |
| pendigits | 88.59 | 77.50 |
| Pima | 68.08 | 72.30 |
| satellite | 69.78 | 86.90 |
| satimage-2 | 99.24 | 99.70 |
| SpamBase | 55.63 | 56.30 |
| thyroid | 95.08 | 94.10 |
| Waveform | 64.14 | 84.30 |
| WDBC | 98.54 | 98.40 |
| AVG | 73.96 | 78.93 |
| AUPRC | | |
| breastw | 97.73 | 96.90 |
| Cardiotocography | 43.52 | 35.30 |
| fault | 35.17 | 49.10 |
| InternetAds | 30.00 | 32.80 |
| landsat | 20.21 | 28.10 |
| optdigits | 2.73 | 5.00 |
| PageBlocks | 43.68 | 46.20 |
| pendigits | 8.47 | 6.00 |
| Pima | 49.18 | 52.90 |
| satellite | 70.21 | 79.20 |
| satimage-2 | 92.35 | 95.70 |
| SpamBase | 40.96 | 42.50 |
| thyroid | 29.17 | 44.30 |
| Waveform | 4.44 | 9.60 |
| WDBC | 52.35 | 57.80 |
| AVG | 41.34 | 45.43 |

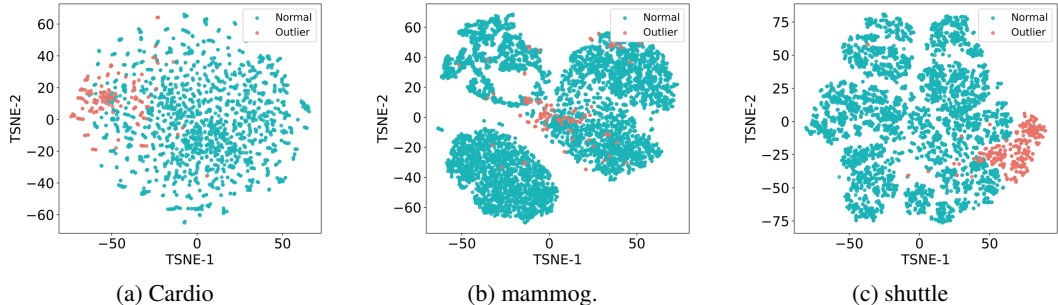

(a) Cardio  (b) mammog.  (c) shuttle

Figure 5: T-SNE visualization results of learned representations $\mathbf{Z}_{T_i}$ on several datasets.

Table 15: Performance (AUROC %) of UniOD on datasets from different domains when removing historical datasets from the corresponding domains.

| | Physical | Image | | | | Astronautics | |
|---|---|---|---|---|---|---|---|
| | magic.gamma | ALOI | celeba | letter | skin | shuttle | AVG |
| UniOD | 70.10 | 54.39 | 81.21 | 64.05 | 76.12 | 99.51 | 78.52 |
| UniOD without Physical | 68.47 | 54.13 | 86.48 | 75.15 | 56.30 | 87.49 | 76.42 |
| UniOD without Astronautics | 68.67 | 54.09 | 83.91 | 70.10 | 62.99 | 98.48 | 76.02 |
| UniOD without Image | 68.44 | 54.65 | 80.94 | 60.89 | 71.05 | 96.85 | 75.32 |

## H.10 EVALUATION OF UNIOD ON LARGE-SCALE DATASETS

In this subsection, we evaluate the performance of UniOD on large-scale datasets. Due to the GPU memory constraint, we randomly partition large-scale datasets into disjoint subsets and run UniOD independently on each partition. Results in Table 16 and Table 17 demonstrate its effectiveness.

Table 16: AUROC comparison on large-scale datasets.

| Data (samples) | kNN | IF | ECOD | DTE-NP | UniOD |
|---|---|---|---|---|---|
| Campaign (41188) | 74.2 | 70.3 | 76.9 | 74.0 | 75.1 |
| shuttle (49097) | 65.8 | 99.7 | 99.2 | 62.5 | 99.2 |

Table 17: AUPRC comparison on large-scale datasets.

| Data (samples) | kNN | IF | ECOD | DTE-NP | UniOD |
|---|---|---|---|---|---|
| Campaign (41188) | 27.7 | 30.7 | 35.4 | 26.6 | 28.9 |
| shuttle (49097) | 17.4 | 97.8 | 90.4 | 15.6 | 93.3 |

## H.11 EXPERIMENTAL RESULTS ON THE OTHER 27 DATASETS FROM ADBENCH

To provide a more comprehensive evaluation, we have now conducted additional experiments using Group I for training and testing on the remaining 27 datasets. As shown in Table 18 and Table 19, the results consistently demonstrate the effectiveness of UniOD.

Table 18: AUROC (%) comparison on the additional 27 datasets of ADBench.

| Dataset | DTE-NP | ECOD | IF | KNN | UniOD |
|---|---|---|---|---|---|
| annthyroid | 81.8 | 79.1 | 81.5 | 79.1 | 68.8 |
| backdoor | 82.5 | 84.2 | 76.6 | 85.4 | 88.3 |
| census | 68.1 | 67.0 | 57.2 | 68.1 | 67.8 |
| donors | 81.6 | 86.4 | 76.3 | 83.5 | 89.1 |
| fraud | 99.5 | 99.4 | 99.1 | 99.6 | 99.2 |
| ionosphere | 92.7 | 72.8 | 84.7 | 92.1 | 81.5 |
| mnist | 82.6 | 74.1 | 79.7 | 85.3 | 86.1 |
| musk | 26.6 | 95.6 | 99.9 | 85.8 | 100.0 |
| stamps | 75.9 | 87.6 | 89.1 | 87.6 | 93.7 |
| yeast | 38.6 | 44.4 | 40.5 | 40.1 | 38.8 |
| CIFAR10 | 85.7 | 84.7 | 85.3 | 88.5 | 89.8 |
| MVTec-AD | 99.4 | 97.9 | 98.5 | 99.3 | 99.0 |
| agnews | 62.5 | 50.5 | 53.0 | 61.3 | 56.0 |
| 20news | 74.2 | 60.4 | 65.0 | 72.2 | 55.0 |
| SVHN | 61.4 | 51.2 | 53.9 | 59.0 | 67.0 |
| MNIST-C | 37.5 | 34.6 | 35.6 | 41.3 | 54.0 |
| FashionMNIST | 81.5 | 87.8 | 87.7 | 85.9 | 41.0 |
| amazon | 55.2 | 51.4 | 48.6 | 54.5 | 89.3 |
| yelp | 58.5 | 56.0 | 51.7 | 58.0 | 64.1 |
| imdb | 51.4 | 47.0 | 49.7 | 51.3 | 50.9 |
| glass | 86.6 | 70.5 | 78.4 | 86.7 | 79.4 |
| Hepatitis | 68.7 | 73.9 | 72.8 | 75.0 | 83.8 |
| Lymphography | 99.4 | 99.5 | 100.0 | 99.8 | 98.1 |
| vertebral | 36.1 | 42.0 | 36.8 | 35.4 | 28.6 |
| WBC | 98.8 | 99.4 | 99.4 | 98.4 | 99.2 |
| wine | 46.0 | 73.3 | 73.7 | 72.8 | 99.6 |
| WPBC | 49.6 | 48.1 | 48.3 | 52.9 | 55.1 |
| Average | 69.7 | 71.1 | 71.2 | 74.0 | 74.9 |

Table 19: AUPRC (%) comparison on the additional 27 datasets of ADBench.

| Dataset | DTE-NP | ECOD | IF | KNN | UniOD |
|---|---|---|---|---|---|
| annthyroid | 23.9 | 27.8 | 31.8 | 23.1 | 21.7 |
| backdoor | 42.7 | 8.6 | 5.1 | 45.5 | 35.4 |
| census | 9.1 | 8.6 | 6.6 | 8.9 | 8.8 |
| donors | 17.3 | 23.3 | 11.5 | 16.9 | 22.5 |
| fraud | 15.5 | 17.9 | 14.7 | 18.6 | 17.7 |
| ionosphere | 92.8 | 64.6 | 79.4 | 91.7 | 64.5 |
| mnist | 38.1 | 17.5 | 25.7 | 40.8 | 39.5 |
| musk | 8.7 | 49.2 | 97.9 | 38.1 | 100.0 |
| stamps | 21.3 | 31.4 | 30.8 | 30.2 | 45.0 |
| yeast | 28.9 | 33.3 | 30.4 | 29.9 | 29.1 |
| CIFAR10 | 42.3 | 31.5 | 31.6 | 51.2 | 46.5 |
| MVTec-AD | 97.7 | 95.3 | 95.9 | 97.6 | 97.4 |
| agnews | 7.2 | 5.0 | 5.4 | 7.0 | 5.8 |
| 20news | 12.0 | 6.6 | 7.2 | 10.7 | 5.7 |
| SVHN | 6.7 | 5.2 | 6.0 | 6.2 | 7.8 |
| MNIST-C | 3.8 | 3.5 | 3.6 | 4.0 | 5.5 |
| FashionMNIST | 22.7 | 32.8 | 34.3 | 30.5 | 4.0 |
| amazon | 5.6 | 5.2 | 4.9 | 5.4 | 38.8 |
| yelp | 6.2 | 5.4 | 5.0 | 6.1 | 7.3 |
| imdb | 5.0 | 4.5 | 4.8 | 4.9 | 5.0 |
| glass | 16.9 | 18.6 | 15.1 | 16.0 | 11.3 |
| Hepatitis | 23.8 | 29.2 | 26.9 | 31.5 | 58.2 |
| Lymphography | 85.6 | 89.7 | 100.0 | 94.8 | 71.4 |
| vertebral | 9.4 | 10.7 | 9.5 | 9.3 | 8.5 |
| WBC | 84.2 | 90.3 | 94.7 | 76.1 | 87.1 |
| wine | 8.0 | 19.1 | 17.4 | 13.8 | 94.3 |
| WPBC | 22.6 | 21.8 | 22.1 | 23.9 | 25.6 |
| Average | 28.1 | 28.0 | 30.3 | 30.9 | 35.7 |

