# OpenReview forum: "UniOD: A Universal Model for Outlier Detection across Diverse Domains"
_ICLR.cc/2026/Conference — ICLR 2026 Poster_

### Official Review · Reviewer_v82N · 2025-10-29

**Soundness:** 2
**Presentation:** 3
**Contribution:** 3
**Rating:** 6
**Confidence:** 4

**Summary:**

The paper proposes UniOD, a universal outlier detection framework that trains a single model on a collection of labeled historical datasets to detect outliers in new, unseen tabular datasets without retraining or hyperparameter tuning. To achieve cross-domain generalization, UniOD constructs multi-scale similarity matrices from each dataset, factorizes them via SVD to obtain uniform-dimensional node features, and then employs a hybrid architecture of GINs and graph transformers to perform node-level binary classification (inlier vs. outlier). The method is evaluated on 15 datasets from ADBench.

**Strengths:**

1.	The proposed method introduces a novel paradigm shift from dataset-specific to universal outlier detection, which is underexplored in the literature.
2.	The methodology is well-motivated and technically sound. The integration of SVD-based feature unification, graph construction, and GNNs is carefully designed to handle varying feature dimensions and semantics. The theoretical analysis provides a nontrivial generalization bound that aligns with empirical findings.
3.	The paper is generally well-written, with clear figures and a logical flow from problem statement to evaluation.

**Weaknesses:**

1.	The authors state that experiments are conducted on 30 datasets from ADBench, split into two groups of 15 for cross-validation. However, ADBench actually contains 57 datasets, not just 30. The paper does not justify why only a subset was selected, nor does it clarify the criteria for partitioning. This raises concerns about potential cherry-picking. The authors should either (a) use the full ADBench tabular benchmark, or (b) explicitly state the selection rationale and provide results on a broader set to demonstrate robustness.
2.	The paper partitions the 30 ADBench datasets into two fixed groups of 15 for training and testing, and performs a single cross-validation swap. While this demonstrates basic robustness, it does not adequately address how historical datasets should be chosen in practice or how performance varies under different selection strategies.
3.	The claim of universality is compelling but narrowly validated. All experiments are on tabular data; it remains unclear whether UniOD can generalize to other modalities (e.g., images, time series) without significant architectural changes. Clarifying the scope of “universal” (i.e., universal across tabular domains only) would temper overstatement.

**Questions:**

Could the authors clarify the criteria used to select these 30 datasets?
Can the authors provide qualitative or quantitative analysis of when and why UniOD fails?

---

> ### Author Response · Authors · 2025-11-21
> **Author Response to Reviewer v82N (1/2)**
>
> Dear Reviewer v82N,
>
> Thank you for reviewing our paper. Your recognition of its novelty and contribution means a great deal to us. We also include all the additional experimental results during rebuttal in Appendix I of our paper. Please refer to it. Our responses are as follows.
>
> **Response to W1**
>
> Thank you for this insightful question. In our initial experiments, the random split of 30 datasets into two groups was primarily motivated by simplicity. To provide a more comprehensive evaluation, we have now conducted additional experiments using Group I for training and testing on the remaining 27 datasets. As shown in the table below, the results consistently demonstrate the effectiveness of UniOD.
>
> **Table 1.** AUROC (\%) comparison on 27 datasets.
>
> | Dataset       | DTE-NP | ECOD | IF   | KNN  | UniOD | Dataset      | DTE-NP | ECOD | IF   | KNN  | UniOD |
> |---------------|--------|------|------|------|-------|--------------|--------|------|------|------|-------|
> | annthyroid    | 81.8   | 79.1 | 81.5 | 79.1 | 68.8  | backdoor     | 82.5   | 84.2 | 76.6 | 85.4 | 88.3  |
> | census        | 68.1   | 67.0 | 57.2 | 68.1 | 67.8  | donors       | 81.6   | 86.4 | 76.3 | 83.5 | 89.1  |
> | fraud         | 99.5   | 99.4 | 99.1 | 99.6 | 99.2  | ionosphere   | 92.7   | 72.8 | 84.7 | 92.1 | 81.5  |
> | mnist         | 82.6   | 74.1 | 79.7 | 85.3 | 86.1  | musk         | 26.6   | 95.6 | 99.9 | 85.8 | 100.0 |
> | stamps        | 75.9   | 87.6 | 89.1 | 87.6 | 93.7  | yeast        | 38.6   | 44.4 | 40.5 | 40.1 | 38.8  |
> | CIFAR10       | 85.7   | 84.7 | 85.3 | 88.5 | 89.8  | MVTec-AD     | 99.4   | 97.9 | 98.5 | 99.3 | 99.0  |
> | agnews        | 62.5   | 50.5 | 53.0 | 61.3 | 56.0  | 20news       | 74.2   | 60.4 | 65.0 | 72.2 | 55.0  |
> | SVHN          | 61.4   | 51.2 | 53.9 | 59.0 | 67.0  | MNIST-C      | 37.5   | 34.6 | 35.6 | 41.3 | 54.0  |
> | FashionMNIST  | 81.5   | 87.8 | 87.7 | 85.9 | 41.0  | amazon       | 55.2   | 51.4 | 48.6 | 54.5 | 89.3  |
> | yelp          | 58.5   | 56.0 | 51.7 | 58.0 | 64.1  | imdb         | 51.4   | 47.0 | 49.7 | 51.3 | 50.9  |
> | glass         | 86.6   | 70.5 | 78.4 | 86.7 | 79.4  | Hepatitis    | 68.7   | 73.9 | 72.8 | 75.0 | 83.8  |
> | Lymphography  | 99.4   | 99.5 | 100.0| 99.8 | 98.1  | vertebral    | 36.1   | 42.0 | 36.8 | 35.4 | 28.6  |
> | WBC           | 98.8   | 99.4 | 99.4 | 98.4 | 99.2  | wine         | 46.0   | 73.3 | 73.7 | 72.8 | 99.6  |
> | WPBC          | 49.6   | 48.1 | 48.3 | 52.9 | 55.1  | **Average**  | 69.7   | 71.1 | 71.2 | 74.0 | 74.9  |
>
>
> **Table 2.** AUPRC (\%) comparison on 27 datasets.
>
> | Dataset       | DTE-NP | ECOD | IF   | KNN  | UniOD | Dataset      | DTE-NP | ECOD | IF   | KNN  | UniOD |
> |---------------|--------|------|------|------|-------|--------------|--------|------|------|------|-------|
> | annthyroid    | 23.9   | 27.8 | 31.8 | 23.1 | 21.7  | backdoor     | 42.7   | 8.6  | 5.1  | 45.5 | 35.4  |
> | census        | 9.1    | 8.6  | 6.6  | 8.9  | 8.8   | donors       | 17.3   | 23.3 | 11.5 | 16.9 | 22.5  |
> | fraud         | 15.5   | 17.9 | 14.7 | 18.6 | 17.7  | ionosphere   | 92.8   | 64.6 | 79.4 | 91.7 | 64.5  |
> | mnist         | 38.1   | 17.5 | 25.7 | 40.8 | 39.5  | musk         | 8.7    | 49.2 | 97.9 | 38.1 | 100.0 |
> | stamps        | 21.3   | 31.4 | 30.8 | 30.2 | 45.0  | yeast        | 28.9   | 33.3 | 30.4 | 29.9 | 29.1  |
> | CIFAR10       | 42.3   | 31.5 | 31.6 | 51.2 | 46.5  | MVTec-AD     | 97.7   | 95.3 | 95.9 | 97.6 | 97.4  |
> | agnews        | 7.2    | 5.0  | 5.4  | 7.0  | 5.8   | 20news       | 12.0   | 6.6  | 7.2  | 10.7 | 5.7   |
> | SVHN          | 6.7    | 5.2  | 6.0  | 6.2  | 7.8   | MNIST-C      | 3.8    | 3.5  | 3.6  | 4.0  | 5.5   |
> | FashionMNIST  | 22.7   | 32.8 | 34.3 | 30.5 | 4.0   | amazon       | 5.6    | 5.2  | 4.9  | 5.4  | 38.8  |
> | yelp          | 6.2    | 5.4  | 5.0  | 6.1  | 7.3   | imdb         | 5.0    | 4.5  | 4.8  | 4.9  | 5.0   |
> | glass         | 16.9   | 18.6 | 15.1 | 16.0 | 11.3  | Hepatitis    | 23.8   | 29.2 | 26.9 | 31.5 | 58.2  |
> | Lymphography  | 85.6   | 89.7 | 100.0| 94.8 | 71.4  | vertebral    | 9.4    | 10.7 | 9.5  | 9.3  | 8.5   |
> | WBC           | 84.2   | 90.3 | 94.7 | 76.1 | 87.1  | wine         | 8.0    | 19.1 | 17.4 | 13.8 | 94.3  |
> | WPBC          | 22.6   | 21.8 | 22.1 | 23.9 | 25.6  | **Average**  | 28.1   | 28.0 | 30.3 | 30.9 | 35.7  |

---

> ### Author Response · Authors · 2025-11-21
> **Author Response to Reviewer v82N (2/2)**
>
> **Response to W2**
>
> Thank you for raising this insightful question. In our ablation study, we investigate how the number of historical datasets influences the performance of UniOD, finding that incorporating more historical datasets consistently improves generalization. This trend aligns with the theoretical generalization bound $\mathcal{L}(f)-\hat{\mathcal{L}}(f)\leq \mathcal{O}(\frac{\ln{M}}{M}+\frac{1}{\sqrt{M}})$ mentioned in Eq.(12) of our paper, where $M$ is the number of historical datasets. In general, using more historical datasets leads to a tighter bound and improved performance. Therefore, in practice, we recommend utilizing as many historical datasets as possible instead of selecting several of them for training.
>
> To further assess UniOD’s robustness to dataset variability, we conducted additional experiments in which UniOD was evaluated on datasets from the physical, astronautics, and image domains after removing all historical datasets from the same domain during training. We observed that excluding domain-specific historical datasets does not lead to a substantial performance drop on the test sets from corresponding domains, indicating that UniOD can generalize effectively to previously unseen domains.
>
> **Table 3.**  Performance (AUROC %) of UniOD on datasets from different domains when removing historical datasets from the corresponding domains.
>
> | Domain             | Physical | Image | Image | Image | Image | Astronautics |     |
> |--------------------|----------|-------|-------|-------|-------|--------------|-----|
> | Method / Dataset   | magic.gamma | ALOI | celeba | letter | skin | shuttle | AVG |
> | UniOD              | 70.10   | 54.39 | 81.21 | 64.05 | 76.12 | 99.51 | 78.52 |
> | UniOD without Physical | 68.47   | 54.13 | 86.48 | 75.15 | 56.30 | 87.49 | 76.42 |
> | UniOD without Astronautics | 68.67 | 54.09 | 83.91 | 70.10 | 62.99 | 98.48 | 76.02 |
> | UniOD without Image    | 68.44   | 54.65 | 80.94 | 60.89 | 71.05 | 96.85 | 75.32 |
>
> Regarding the selection of historical datasets, we agree that identifying which datasets contribute most effectively is an important direction. In future work, we plan to evaluate the contribution of each historical dataset and explore principled strategies for selecting informative historical datasets.
>
> **Response to W3**
>
> We thank the reviewer for this important remark. Our primary focus in the current paper is on tabular datasets drawn from diverse domains, and we will clarify the intended scope of “universal” in the revised manuscript.
>
> Nevertheless, data from other modalities can often be converted into tabular form using pretrained feature extractors (e.g., ViT or CLIP for images, and BERT or other LLMs for sequential or natural language data). Once represented in this way, UniOD can be applied to these modalities without any architectural changes.
>
> Moreover, in our extended evaluation on 27 additional datasets, UniOD achieves strong performance on 10 datasets whose inputs are feature-extracted representations from the image and NLP domains, including CIFAR10, FashionMNIST, MNIST-C, MVTec-AD, SVHN (image) and Agnews, Amazon, IMDB, Yelp, 20Newsgroups (text). These results suggest that the proposed framework can be effectively used on non-tabular modalities via standard feature extraction.
>
> **Response to Q1**
>
> Thank you for these insightful questions. As mentioned in our response to W2, the initial set of 30 datasets was randomly selected from ADBench for evaluation. During rebuttal, we additionally include results on the remaining 27 datasets to provide a more complete assessment.
>
> Regarding the failure cases of UniOD, we observe that simple methods such as KNN outperform UniOD on several datasets. These datasets typically exhibit either very high anomaly ratios or high dimensionality. Both factors can affect UniOD’s performance for the following reasons:
> (i) High anomaly ratio. Most historical datasets used for training UniOD have anomaly ratios below 10\%. When UniOD is applied to datasets with substantially higher anomaly ratios (e.g., Cardio (22\%) and Pima (34\%)), the mismatch between training and target anomaly ratios may lead to underperformance. A simple remedy is to subsample historical datasets to create variants with varying anomaly ratios.
> (ii) High dimensionality. It is well known that Euclidean distances become less informative in high-dimensional spaces. Since UniOD constructs similarity matrices based on Euclidean distances, its effectiveness may degrade on datasets with very high dimensionality. Exploring alternative similarity measures or dimensionality reduction priors to mitigate this limitation is our future work direction.
>
> ****Thank you again for your time and effort in reviewing our paper along with this rebuttal. We are looking forward to your feedback, and please do not hesitate to let us know if there are any concerns or questions still not properly addressed.****

---

### Official Review · Reviewer_efRk · 2025-10-30

**Soundness:** 3
**Presentation:** 3
**Contribution:** 3
**Rating:** 6
**Confidence:** 4

**Summary:**

This paper proposes UniOD, a universal outlier detection model designed to address the core limitations of conventional methods, including the significant efforts required for hyperparameter tuning, the need for repetitive model training on new datasets, and the inability to leverage knowledge from historical datasets. The key contributions of this work are fourfold:

1. **Problem Formulation**: It is the first to explicitly define and formalize the problem of universal outlier detection, which aims to "train a single model that can directly generalize to any unseen tabular datasets from diverse domains without any retraining or hyperparameter tuning".

2. **Technical Framework**: It introduces a novel technical framework based on a "graph reformulation" process, which unifies heterogeneous datasets by constructing multi-scale similarity matrices, decomposing them into uniformly dimensioned features via SVD, and subsequently employing graph neural networks for node classification.

3. **Theoretical Analysis**: It provides a theoretical generalization bound, offering mathematical guarantees for the effectiveness of the proposed method.

4. **Experimental Validation**: Extensive experiments demonstrate that UniOD outperforms 17 baseline methods on 30 datasets while achieving faster inference speed.

**Strengths:**

This paper presents several noteworthy strengths:

1. **Paradigm-Shifting Contribution**
   - First explicit proposal and systematic formalization of "universal outlier detection"
   - Core innovation: single model generalizing across diverse domains without retraining or hyperparameter tuning
   - Represents fundamental reformulation of conventional outlier detection paradigm

2. **Technical Innovation**
   - Elegant graph reformulation process transforms heterogeneous tabular data
   - Creates unified, structurally learnable representations from disparate datasets
   - Overcomes limitations of existing transfer learning approaches that require:
     - Extensive hyperparameter evaluation
     - Strong domain similarity assumptions

3. **Theoretical Rigor**
   - Provides generalization error analysis beyond empirical demonstrations
   - Offers mathematical guarantees for method effectiveness
   - Enhances academic credibility and theoretical foundation

**Weaknesses:**

While the proposed UniOD framework demonstrates compelling performance, several limitations warrant discussion for future improvement:

1. **Scalability Challenges in Preprocessing**
The O(n²) computational and memory requirements for similarity matrix construction present practical constraints, as evidenced by the needed subsampling for datasets beyond 6,000 samples. Future work could explore approximate nearest neighbor techniques or sparse graph construction to enhance applicability to larger-scale datasets.

2. **Dependence on Euclidean-Based Similarity**
The reliance on Gaussian kernel similarity (and consequently Euclidean distance) may limit performance in scenarios where this metric is suboptimal. Incorporating learnable or adaptive distance metrics could strengthen robustness across diverse data distributions.

3. **Sensitivity to Historical Data Availability**
While leveraging multiple labeled historical datasets is a strength, the framework's effectiveness in domains with extreme scarcity of labeled anomalies remains unverified. Investigating few-shot or semi-supervised adaptations would valuablely expand its applicability.

**Questions:**

1. **Scalability and Computational Efficiency**
The paper indicates that datasets exceeding 6,000 samples required subsampling due to computational constraints. Have the authors explored more scalable graph construction alternatives, such as k-nearest neighbor sparse graphs or Nyström approximation methods, to avoid full similarity matrix computation? Could experimental results demonstrate whether these approaches maintain performance while handling datasets at larger scales (e.g., tens of thousands of samples), thereby improving practical applicability in big data scenarios?
2. **Analysis of Performance Boundaries**
While UniOD shows strong average performance, simpler methods outperform it on specific datasets (e.g., Cardiotocography, Pima). Could the authors provide further analysis identifying dataset characteristics where UniOD may underperform? For instance, are there correlations with meta-features like anomaly ratio, dimensionality, or cluster structure clarity? Defining such boundaries would offer valuable guidance for practical application.

---

> ### Author Response · Authors · 2025-11-21
> **Author Response to Reviewer efRK (1/2)**
>
> Dear Reviewer efRk,
>
> We are very grateful for your recognition of the novelty and contribution of our method, and we appreciate your suggestions for improving our work.  We also include all the additional experimental results during rebuttal in Appendix I of our paper. Please refer to it. Our responses are as follows.
>
>
> **Response to W1**
>
> We thank the reviewer for raising the important point regarding the computational and memory overhead of constructing similarity matrices. The suggestion to employ alternatives like k-nearest-neighbor sparse graphs or the Nyström approximation is excellent, and we will certainly explore these directions in future work.
>
> To clarify, the primary driver for subsampling is the GPU memory constraint during training and inference. Our framework constructs $K=5$ dense graphs per dataset, processed by 5 GINs and GTs. Since embeddings from all layers must be stored, memory consumption becomes substantial. To address this bottleneck, a practical solution is to (randomly) partition large-scale datasets into disjoint subsets and run UniOD independently on each partition. We have evaluated this approach on two large datasets. The results, shown in the following table, demonstrate its effectiveness.
>
> **Table 1.** AUROC comparison on large scale datasets.
>
> | Data (samples)     | kNN  | IF   | ECOD | DTE-NP | UniOD |
> |--------------------|------|------|------|--------|-------|
> | Campaign (41188)   | 74.2 | 70.3 | 76.9 | 74.0   | 75.1  |
> | shuttle (49097)    | 65.8 | 99.7 | 99.2 | 62.5   | 99.2  |
>
> **Table 2.** AUPRC comparison on large scale datasets.
>
> | Data (samples)     | kNN  | IF   | ECOD | DTE-NP | UniOD |
> |--------------------|------|------|------|--------|-------|
> | Campaign (41188)   | 27.7 | 30.7 | 35.4 | 26.6   | 28.9  |
> | shuttle (49097)    | 17.4 | 97.8 | 90.4 | 15.6   | 93.3  |
>
>
> **Response to W2**
>
> Thank you for this constructive suggestion. We agree that exploring learnable or adaptive distance metrics, as well as alternative graph construction methods such as the Laplacian kernel, kNN graph, and $\epsilon$-neighborhood graph, would be an interesting direction for future work. Here, we provide a brief discussion of these choices.
>
> Compared with the Gaussian kernel used in our method:
> (a) The Laplacian kernel is generally more robust to outliers, which may cause the resulting graph to suppress or overlook subtle structural information associated with rare but important anomalous points.
> (b) The kNN graph and $\epsilon$-neighborhood graph tend to preserve less global information and might be unconnected if $k$ and $\epsilon$ are too small.
>
> **Response to W3**
>
> Thank you for highlighting the importance of evaluating the model’s effectiveness in scenarios with extremely limited labeled anomalies or entirely unseen domains. To address this concern, we conducted additional experiments in which UniOD was tested on datasets from the physical, astronautics, and image domains after removing all historical datasets from the same domain during training. excluding domain-specific historical datasets does not cause a substantial performance degradation on the corresponding test sets, suggesting that UniOD can generalize effectively to previously unseen domains.
>
> We attribute this robustness to the fact that UniOD does not rely on raw features. Instead, it construct similarity matrices via Gaussian kernels and generates uniformly dimensioned representations via these   matrices. Consequently, datasets from different domains may still exhibit comparable structural patterns in their similarity matrices, which facilitates effective cross-domain generalization.
>
> We also sincerely appreciate your suggestion regarding few-shot or semi-supervised extensions of UniOD. Exploring these directions is indeed promising, and we plan to investigate them in future work.
>
> **Table 3.** Performance (AUROC %) of UniOD on datasets from different domains when removing historical datasets from the corresponding domains.
>
> | Domain             | Physical | Image | Image | Image | Image | Astronautics |     |
> |--------------------|----------|-------|-------|-------|-------|--------------|-----|
> | Method / Dataset   | magic.gamma | ALOI | celeba | letter | skin | shuttle | AVG |
> | UniOD              | 70.10   | 54.39 | 81.21 | 64.05 | 76.12 | 99.51 | 78.52 |
> | UniOD without Physical | 68.47   | 54.13 | 86.48 | 75.15 | 56.30 | 87.49 | 76.42 |
> | UniOD without Astronautics | 68.67 | 54.09 | 83.91 | 70.10 | 62.99 | 98.48 | 76.02 |
> | UniOD without Image    | 68.44   | 54.65 | 80.94 | 60.89 | 71.05 | 96.85 | 75.32 |

---

> > ### Author Response · Authors · 2025-11-21
> > **Author Response to Reviewer efRK (2/2)**
> >
> > **Response to Q1**
> >
> > Thanks again for your valuable suggestions. Please refer to response to W1.
> >
> > **Response to Q2**
> >
> > Thank you for this insightful question regarding the performance boundaries of UniOD. Our analysis has identified two key meta-features that influence its performance: anomaly ratio and data dimensionality.
> >
> > Anomaly Ratio: UniOD is primarily trained on historical datasets where the anomaly ratio is typically below 10\%. Consequently, when encountering datasets with a significantly higher anomaly ratio—such as Cardio (22\%) or Pima (34\%)—its performance may decline. A straightforward strategy to address this limitation is to augment the training data by sub-sampling existing datasets to simulate a wider range of anomaly ratios, thereby enhancing the model's robustness.
> >
> > Dimensionality: As dimensionality increases, the effectiveness of Euclidean distance tends to diminish, a phenomenon well-known in high-dimensional spaces. Since UniOD's similarity matrices are constructed based on Euclidean distance, this can lead to performance decline on high-dimensional datasets.
> >
> > ****Thank you again for your time and effort in reviewing our paper along with this rebuttal. We are looking forward to your feedback, and please do not hesitate to let us know if there are any concerns or questions still not properly addressed.****

---

### Official Review · Reviewer_53cw · 2025-11-01

**Soundness:** 3
**Presentation:** 3
**Contribution:** 2
**Rating:** 6
**Confidence:** 2

**Summary:**

This paper proposed UniOD, a universal pretrained model for outlier detection. UniOD learns from various historical tabular datasets and can directly score the outliers for new datasets in a zero-shot manner, without re-training the model or hyperparameter tuning. The datasets are converted into point-wise kernel matrices at different scales and a GNN is trained on graph-structured datasets.

**Strengths:**

1. Plug-and-play: this paper proposes the UniOD framework by pretraining on various datasets and conduct inference in a zero-shot manner, saving the deployment cost for OD tasks.
2. Unified dataset representation: this paper utilizes the multi-scale similarity and SVD to produce unified node features, enforcing the generalizability of the model.
3. The authors provides both comprehensive theoretical justification and extensive empirical analysis of the method. UniOD is well theoretical grounded.

**Weaknesses:**

1. Dependence on historical datasets. UniOD requires labeled historical datasets, which can be unavailable in real-world applications. The limited historical datasets may impair the performance of UniOD on new datasets, especially when the historical datasets is limited to few domains.
2. Generality concern: the effect of dataset variability is not fully discussed in the paper, as this will potentially influence the model's generality if the model hasn't encountered datasets from similar distributions.

**Questions:**

See the weaknesses above.

---

> ### Author Response · Authors · 2025-11-21
>
> Dear Reviewer 53cw,
>
> We sincerely appreciate your recognition of the effectiveness of our method and are grateful for your suggestions that improved our work. We also include all the additional experimental results during rebuttal in Appendix I of our paper. Please refer to it. Our responses are as follows.
>
> **Response to W1(Dependence on historical datasets)**
>
>
> We thank the reviewer for this insightful question. We would first like to clarify that, in many real-world applications, historical datasets collected from diverse domains are often readily available in today’s data-rich environment. Nevertheless, it is indeed possible that a new test dataset comes from a previously unseen domain, and we have therefore conducted additional experiments to directly address this concern. Specifically, we evaluated UniOD on datasets from the physical, astronautics, and image domains after removing all historical datasets belonging to the same domain during training. The results show that excluding domain-specific datasets does not lead to a substantial performance drop on the corresponding test domain, suggesting that UniOD exhibits strong generalization to previously unseen domains.
>
> We attribute this generalization ability to the fact that UniOD does not rely on raw input features. Instead, it constructs uniformly dimensioned representations through similarity matrices. As a result, even datasets from very different domains can share similar structural patterns in their similarity matrices, which enables effective cross-domain generalization.
>
> **Table.** Performance (AUROC %) of UniOD on datasets from different domains when removing historical datasets from the corresponding domains.
>
> | Domain                | Physical | Image | Image | Image | Image | Astronautics |     |
> |-----------------------|----------|-------|-------|-------|-------|--------------|-----|
> | Method / Dataset      | magic.gamma | ALOI | celeba | letter | skin | shuttle | AVG |
> | UniOD                 | 70.10   | 54.39 | 81.21 | 64.05 | 76.12 | 99.51 | 78.52 |
> | UniOD without Physical    | 68.47   | 54.13 | 86.48 | 75.15 | 56.30 | 87.49 | 76.42 |
> | UniOD without Astronautics| 68.67   | 54.09 | 83.91 | 70.10 | 62.99 | 98.48 | 76.02 |
> | UniOD without Image       | 68.44   | 54.65 | 80.94 | 60.89 | 71.05 | 96.85 | 75.32 |
>
> **Response to W2 (Generality concern)**
>
> We thank the reviewer for raising this concern. As noted in our response to W1, the additional experiments show that removing domain-specific historical datasets does not lead to a significant performance decline for UniOD. Together with the cross-validation results, this suggests that UniOD is not overly sensitive to the specific domain composition of the historical datasets.
>
> We agree, however, that UniOD may underperform on test datasets whose distributions differ substantially from those of the historical datasets. Importantly, such distributional differences are not solely determined by the domain or feature space, since each dataset is ultimately converted into graphs using a Gaussian kernel. Instead, factors such as anomaly ratio, dimensionality, and other meta-properties tend to play a more critical role. For example, we observe that UniOD underperforms on several datasets with high anomaly ratios (e.g., Cardio, Pima), which we attribute primarily to the scarcity of high–anomaly–ratio datasets in the historical training pool. A simple remedy is to subsample the historical datasets to create variants with different anomaly ratios, thereby enriching the diversity of training conditions and further improving UniOD’s generality on datasets with high anomaly ratios.
>
> In our ablation study, we also investigate how the number of historical datasets influences the performance of UniOD, which is closely related to dataset-level variability. The results indicate that using more historical datasets tends to yield better generalization performance, which is consistent with the theoretical generalization bound $\mathcal{L}(f)-\hat{\mathcal{L}}(f)\leq \mathcal{O}\left(\frac{\ln M}{M}+\frac{1}{\sqrt{M}}\right)$ given in Eq.(12) of our paper, where $M$ denotes the number of historical datasets. Regarding more fine-grained forms of variability, such as dataset-specific variability, systematically analyzing this variability and quantifying the contribution or importance of each historical dataset is an interesting and important direction for future work.
>
> **Thank you for your time and effort in reviewing our paper along with this rebuttal. We are looking forward to your feedback, and please do not hesitate to let us know if there are any concerns or questions still not properly addressed.**

---

> > ### Comment · Reviewer_53cw · 2025-11-26
> >
> > The authors' response addresses most of my concerns. I recommend acceptance. Increased my score to 8.

---

> > > ### Author Response · Authors · 2025-11-26
> > >
> > > Dear Reviewer 53cw,
> > >
> > > We are very grateful for your feedback.

---

### Official Review · Reviewer_tKHD · 2025-11-01

**Soundness:** 3
**Presentation:** 4
**Contribution:** 3
**Rating:** 6
**Confidence:** 4

**Summary:**

This paper introduces UniOD, a novel framework for outlier detection that is generalizable to diverse datasets. The authors address the inefficiency of conventional OD methods, which typically require per-dataset retraining and hyperparameter tuning. UniOD tackles this by leveraging a collection of historical labeled datasets to train a single, universal model. The framework's key innovation is its data unification pipeline, which transforms each dataset into a set of multi-scale similarity matrices to create graph-structured representations. It then employs Singular Value Decomposition (SVD) to generate uniformly dimensioned features. Outlier detection is thus reformulated as a node classification task on these graphs, tackled by a GNN-based model. Once trained, UniOD can be directly applied to new, unseen datasets without any further training, demonstrating superior performance against 17 baseline methods on a benchmark of 30 datasets.

**Strengths:**

•	A Novel Paradigm for OD: The primary strength is the innovative concept of a universal framework that eliminates the need for per-dataset retraining. This directly addresses a major bottleneck in the practical application of outlier detection.
•	Elegant Unification of Heterogeneous Data: The use of multi-scale similarity matrices combined with SVD is a powerful and clever technique for creating a unified feature space from datasets with diverse dimensionalities and semantics.
•	Strong Empirical and Theoretical Backing: The claims are convincingly supported by comprehensive experiments on a large benchmark, which is further bolstered by a theoretical analysis of the model's generalization ability.
•	High Practicality and Efficiency: By decoupling training from testing, the framework is highly practical and computationally efficient at inference time, making it well-suited for real-world scenarios requiring rapid analysis of new data.

**Weaknesses:**

•	Heavy Reliance on Historical Data Composition: The model's success is fundamentally tied to the quality, scale, and diversity of the historical datasets. The paper lacks an investigation into the sensitivity of the model to the composition of this training pool.
•	Potential Scalability Bottlenecks: The methodology relies on constructing an n*n similarity matrix, which has a quadratic complexity (O(n²)) with respect to the number of samples. This could be computationally prohibitive for very large datasets.
•	Limited Exploration of Graph Construction: The framework exclusively uses a Gaussian kernel to build the similarity matrices. An investigation into different kernel functions or alternative graph construction techniques would have strengthened the paper's claims of robustness.

**Questions:**

1.	It will strengthen the paper if providing an analysis of the model's sensitivity to the composition of the historical training data. For example, how does performance on a specific target domain (e.g., finance) change when all finance-related datasets are deliberately excluded from the training pool? This would clarify the practical requirements for curating the training set.
2.	The O(n²) complexity for similarity matrix construction is a potential bottleneck. It will be helpful to explore more scalable graph construction techniques, such as those based on approximate nearest neighbors, to enhance the framework's applicability to datasets with millions of samples.
3.	It will be helpful to provide some qualitative analysis on the structural patterns the GNN model learns to distinguish outliers. For instance, in the graph representation, are outliers typically identified as isolated nodes, or do they belong to small, dense clusters disconnected from the main graph component? This would provide valuable insight into the model's decision-making process.

---

> ### Author Response · Authors · 2025-11-21
> **Author Response to Reviewer tKHD (1/2)**
>
> Dear Reviewer TKHD,
>
> We are sincerely grateful for your appreciation for novelty and effectiveness of our method. Thanks for your constructive suggestions. We also include all the additional experimental results during rebuttal in Appendix I of our paper. Please refer to it. Our responses are as follows.
>
> **Response to W1 (Historical Data Composition)**
>
> We thank the reviewer for highlighting the importance of analyzing the model’s sensitivity to the composition of the historical training data. To address this concern, we conducted additional experiments where UniOD is evaluated on datasets from the physical, astronautics, and image domains, while systematically removing all historical datasets belonging to the same domain or field during training. The results are shown in the following table. We observe that excluding these domain-specific datasets does not lead to a significant performance drop on the corresponding test domain, suggesting that UniOD is not overly sensitive to the particular composition of the historical training data.
>
> We attribute this robustness to two main factors:
>    (i) Even among tabular datasets within the same domain, the feature spaces and data characteristics can vary substantially.
>    (ii) UniOD does not directly rely on the original raw features. Instead, it leverages similarity matrices to construct uniformly dimensioned representations across datasets. As a result, datasets from different domains may still exhibit similar structural patterns in their similarity matrices, enabling effective cross-domain generalization. Therefore, for a domain without historical data, our model, learned from the historical data of other domains, performs well.
>
> **Table:** Performance (AUROC \%) of UniOD on datasets from different domains when removing historical datasets from the corresponding domains.
>
> | Domain              | Physical | Image | Image | Image | Image | Astronautics |      |
> |---------------------|----------|-------|-------|-------|-------|--------------|------|
> |  | magic.gamma | ALOI | celeba | letter | skin | shuttle | AVG |
> | UniOD               | 70.10   | 54.39 | 81.21 | 64.05 | 76.12 | 99.51 | 78.52 |
> | UniOD without Physical  | 68.47   | 54.13 | 86.48 | 75.15 | 56.30 | 87.49 | 76.42 |
> | UniOD without Astronautics | 68.67| 54.09 | 83.91 | 70.10 | 62.99 | 98.48 | 76.02 |
> | UniOD without Image     | 68.44   | 54.65 | 80.94 | 60.89 | 71.05 | 96.85 | 75.32 |
>
> **Response to W2 (Potential Scalability Bottlenecks)**
>
> We thank the reviewer for this insightful question. We agree that constructing the full similarity matrix can be computationally expensive for large-scale datasets. A practical and well-established solution is to employ Nyström approximation methods, which avoid computing the entire matrix. By selecting $m \ll n$ Nyström landmark points, the computational cost can be reduced to $\mathcal{O}(mn)$, making the construction of the similarity matrix substantially more efficient in practice. Another simple strategy to address this bottleneck is to (randomly) partition large-scale datasets into disjoint subsets and run UniOD independently on each partition. We have evaluated this approach on two large datasets. The results, shown in the following table, demonstrate its effectiveness.
>
> **Table.** AUROC comparison on large scale datasets.
>
> | Data (samples)     | kNN  | IF   | ECOD | DTE-NP | UniOD |
> |--------------------|------|------|------|--------|-------|
> | Campaign (41188)   | 74.2 | 70.3 | 76.9 | 74.0   | 75.1  |
> | shuttle (49097)    | 65.8 | 99.7 | 99.2 | 62.5   | 99.2  |
>
> **Table.** AUPRC comparison on large scale datasets.
>
> | Data (samples)     | kNN  | IF   | ECOD | DTE-NP | UniOD |
> |--------------------|------|------|------|--------|-------|
> | Campaign (41188)   | 27.7 | 30.7 | 35.4 | 26.6   | 28.9  |
> | shuttle (49097)    | 17.4 | 97.8 | 90.4 | 15.6   | 93.3  |
>
>
> **Response to W3 (Graph Construction)**
>
> Thanks for pointing that out. In our future work, we will consider other graph construction methods such as the Laplacian kernel, kNN graph, and $\epsilon$-neighborhood graph. Here, we'd like to discuss a little bit more about these choices. Compared to the Gaussian kernel used in our paper: a) the Laplacian kernel is more robust to outliers and hence the constructed graph may miss some information of the outliers; b) the kNN graph and $\epsilon$-neighborhood graph preserve little global information of the data.

---

> ### Author Response · Authors · 2025-11-21
> **Author Response to Reviewer tKHD (2/2)**
>
> **Response to Q1**
>
> Please refer to the response for W1.
>
> **Response to Q2**
>
> Please refer to the response for W2.
>
> **Response to Q3**
>
> We appreciate the reviewer’s suggestion regarding visualization. Accordingly, we include t-SNE plots of the learned representations $\mathbf{Z}_{T_i}$ on multiple datasets to illustrate their structure in **Figure 5 from Appendix I.1 of our paper**. We observe that most outliers tend to concentrate into a small, dense cluster, while a smaller portion of outliers appear as isolated points.
>
> We also notice that the separation between normal data and outliers is not particularly pronounced in the t-SNE space, which is likely because $\mathbf{Z}_{T_i}$ are high-dimensional representations (with dimensionality $> 1000$). As such, although a simple MLP can effectively separate normal and anomalous samples in this high-dimensional space, t-SNE may not faithfully preserve the underlying geometry of the original embedding space.
>
> **Thanks for reviewing our paper and this rebuttal. We are looking forward to your feedback and please do not hesitate to let us know if there are any concerns or questions still not properly addressed.**

---

### Author Response · Authors · 2025-12-02
**A Brief Summary of Rebuttal**

Dear Area Chair and Senior Area Chair,

We greatly appreciate the time and effort you have dedicated to reviewing our submission. We fully acknowledge the challenges posed by the API bug in the OpenReview system, which have understandably increased your workload. We are sincerely grateful for your understanding and continued engagement.

In order to facilitate a more streamlined discussion and assist in the final evaluation, we have summarized the main concerns raised by the reviewers along with our responses in the table below. We trust that this summary will provide a clear and convenient overview to support the rebuttal process.

**Reviewer tKHD：**
| NO. | Reviewer Concerns | Our Responses|
|------------------|-----|-------|
|Weaknesses|||
|W1|Model’s sensitivity to the composition of the historical training data.|Additional experiments for evaluating model's sensitivity is provided.|
|W2|Potential scalability bottlenecks for similarity matrix construction.|Several practical solutions are provided with experimental evaluation.|
|W3|Limited exploration of graph construction.| More graph construction will be explored in our future work and reasons for selecting Gaussian kernel is explained.|
|Questions|||
|Q1|Same as W1.|Same as W1.|
|Q2|Same as W2.|Same as W2.|
|Q3|Provide some qualitative analysis on the structural patterns the GNN model learns to distinguish outliers.|t-SNE visualization of learned embedding from GNNs is provided.|

**No reply from the reviewer before OpenReview bug is received.**

**Reviewer 53cw：**
| NO. | Reviewer Concerns | Our Responses|
|------------------|-----|-------|
|Weaknesses|||
|W1|Dependence on historical datasets.| Historical datasets are often readily available and experiments are conducted to evaluate model's robustness to training data.
|W2|More discussion on effect of dataset variability. |Discussion of dataset variability is provided and more specific dataset variability will be explored as future work.|

**We recevied response from the reviewer before the OpenReview bug. The reviewer claimed that most of their concerns have been solved and recommend acceptance (increasing score to 8).**

**Reviewer efRk:**
 NO. | Reviewer Concerns | Our Responses|
|------------------|-----|-------|
|Weaknesses|||
|W1|Scalability Challenges in Preprocessing| Several practical solutions are provided with experimental evaluation.|
|W2|Dependence on Euclidean-Based Similarity| Reasons for selecting Gaussian kernel are explained and more graph construction will be explored in our future work.|
|W3|Sensitivity to Historical Data Availability| Additional experiments for evaluating model's sensitivity are provided.|
|Questions|||
|Q1|Same as W1|Same as W1|
|Q2|Analysis of Performance Boundaries| We provided the failure cases for UniOD and corresponding solutions.|

**No reply from the reviewer before OpenReview bug is received.**

**Reviewer v82N:**
 NO. | Reviewer Concerns | Our Responses|
|------------------|-----|-------|
|Weaknesses|||
|W1|Evaluation on the full ADBench should be provided.|We provide more experimental results on the other 27 datasets of ADBench.|
|W2|More robustnes analysis and practical historical data selection strategy|In practice, one should use as much historical data as possible according to the experimental results and theoretical generalization bound. Additional experiments for evaluating model's robustness are provided.|
|W3|Whether UniOD can generalize to other modalities|experimental results on datasets from other modalities are provided which illustrate generalization ability of UniOD to other modalities.|
|Questions|||
|Q1|Same as W1|Same as W1|
|Q2|Analysis of failure case for UniOD|We analysised the factors lead to failure of UniOD and provided corresponding solutions.|

**No reply from the reviewer before OpenReview bug is received.**

We would like to mention that several common concerns raised by other reviewers can be validated as solved by the reviewer (53cw) engaged in discussion:

 |Common Concerns | Raised By |Validated By 53cw|
|------------------|-----|-------|
|Sensitivity to Historical Data Availability.|tKHD, 53cw, efRk, v82N|"The authors' response addresses most of my concerns."|
|Generality concern|53cw, efRk, v82N|"The authors' response addresses most of my concerns."|

**We sincerely appreciate the dedication, insight, and thoughtful consideration you've devoted to reviewing our work. If you have any questions or require further clarification, please don't hesitate to reach out.**

Sincerely,

Authors

---

### Meta-Review · Area_Chair_zmA9 · 2026-01-04

**Summary:**

In this paper,  the authors consider to build a universal outlier detection model, which is called "UniOD". The basic idea is to use labeled historical datasets to train a universal model capable of detecting outliers for all other tabular datasets without retraining. The proposed method utilizes   uniformly extracted dimensioned features for different datasets. Further, they apply graph neural networks to capture within-dataset and between-dataset information. Finally, the method re-formulate the outlier detection tasks
as node classification tasks.

All the four reviewers give positive scores, and agree that the proposed method UniOD is novel. But there are also some common concerns, such as the scalability and dataset availability. In summary, I believe the paper proposed an interesting work for outlier detection, and conducted sufficient experiments to validate the effectiveness and efficiency.

**Reviewer Concerns:**

The first major concern is about the scalability, which was raised by Reviewer tKHD and efRk. Although the authors explained that this issue can be solved by some fast linear algebra techniques, like Nyström approximation methods, I believe this is still an intrinsic drawback of the proposed method. I suggest the authors may think about how to fundamentally avoid this bottleneck in future.

Second, dataset availability issue raised by Reviewer 53cw and efRk. Although the authors explained this issue from practical perspective, I still think it could be an important problem in some scenarios that may hinder its practicality.

**Reviewer Scores:**

Reviewer tKHD, Reviewer efRk, and Reviewer v82N all give 6. I think they may maintain their scores or slightly increase them.

Reviewer 53cw increased the score from 6 to 8.

---

### Decision · Program_Chairs · 2026-01-26

Accept (Poster)